# Multi-hazard risk assessment for roads: Probabilistic versus deterministic approaches

Stefan Oberndorfer[1,2], Philip Sander[3], Sven Fuchs[2]

[1]Chartered Engineering Consultant for Mountain Risk Engineering and Risk Management, Ecking 57, 5771 Leogang, Austria

[2]Institute of Mountain Risk Engineering, University of Natural Resources and Life Sciences, Peter Jordan Straße 82, 1190 Vienna, Austria

[3]Institute of Construction Management, Bundeswehr University Munich, Werner-Heisenberg-Weg 39, 85577 Neubiberg, Germany

*Correspondence to:* Stefan Oberndorfer (office@oberndorfer-zt.at)

**Abstract.** Mountain hazard risk analysis for transport infrastructure is regularly based on deterministic approaches. Standard risk assessment approaches for roads need a variety of variables and data for risk computation, however without considering potential uncertainty in the input data. Consequently, input data needed for risk assessment is normally processed as discrete mean values without scatter, or as an individual deterministic value from expert judgement if no statistical data is available. To overcome this gap, we used a probabilistic approach to analyse the effect of input data uncertainty on the results, taking a mountain road in the Eastern European Alps as case study. The uncertainty of the input data is expressed with potential bandwidths using two different distribution functions. The risk assessment included risk for persons, property risk and risk for non-operational availability exposed to a multi-hazard environment (torrent processes, snow avalanches, rock fall). The study focuses on the epistemic uncertainty of the risk terms (exposure situations, vulnerability factors, monetary values) ignoring potential sources of variation in the hazard analysis. As a result, reliable quantiles of the calculated probability density distributions attributed to the aggregated road risk due to the impact of multiple-mountain hazards were compared to the deterministic outcome from the standard guidelines on road safety. The results based on our case study demonstrate that with common deterministic approaches risk might be underestimated in comparison to a probabilistic risk modelling setup, mainly due to epistemic uncertainties of the input data. The study provides added value to further develop standardized road safety guidelines and may therefore be of particular importance for road authorities and political decision-makers.

## 1 Introduction

Mountain roads are particularly prone to natural hazards, and consequently, risk assessment for road infrastructure focused on a range of different hazard processes, such as landslides (Benn, 2005; Schlögl et al., 2019), rockfall (Bunce et al., 1997; Hungr and Beckie, 1998; Roberds, 2005; Ferlisi et al., 2012; Michoud et al., 2012; Unterrader et al., 2018) and snow avalanches (Schaerer, 1989; Kristensen et al., 2003; Margreth et al., 2003; Zischg et al., 2005; Hendrikx and Owens, 2008; Rheinberger et al., 2009; Wastl et al., 2011). These studies have in common that they exclusively address the interaction of individual hazards with values at risk of the built environment and/or of society and use qualitative, semi-quantitative and/or quantitative approaches. However, there is still a gap in multi-hazard

risk assessments for road infrastructure. The article provides a comparison of a standard (deterministic) risk
assessment approach for road infrastructure exposed to a multi-hazard environment with a probabilistic risk analysis
method to show the potential bias in the results. The multi-hazard scope of the study is based on a spatially-oriented
approach to include all relevant hazards within our study area. Using this approach, we address the consequences of
multiple hazard impact on road infrastructure and compare the monetary loss of the different hazard types. The
standard framework from ASTRA (2012) for road risk assessment is based on a deterministic approach and
computes road risk based on a variety of input variables. Data is generally addressed with single values without
considering potential input data uncertainty. We used this standardized framework for operational risk assessment for
roads and transportation networks and supplemented this well-established deterministic method with a probabilistic
framework for risk calculation (Fig. 1). A probabilistic approach enables the quantification of epistemic uncertainty
and uses probability distributions to characterize data uncertainty of the input variables while a deterministic
computation uses single values with discrete values without uncertainty representation. While the former calculates
risk with constant or discrete values, ignoring the epistemic uncertainty of the variables, the latter enables the
consideration of the potential range of parameter value by using different distributions to characterize the input data
uncertainty. Our study focuses on the epistemic uncertainty of the risk terms (exposure situations, vulnerability
factors, monetary values) ignoring potential sources of variation within hazard analysis. Thus, the probability of
occurrence of the hazard event was not assessed in a probabilistic way. Since deriving the likelihood of occurrence as
part of the hazard analysis is crucial for risk analysis, a high source of uncertainty is attributed to this factor (Schaub
and Bründl, 2010).

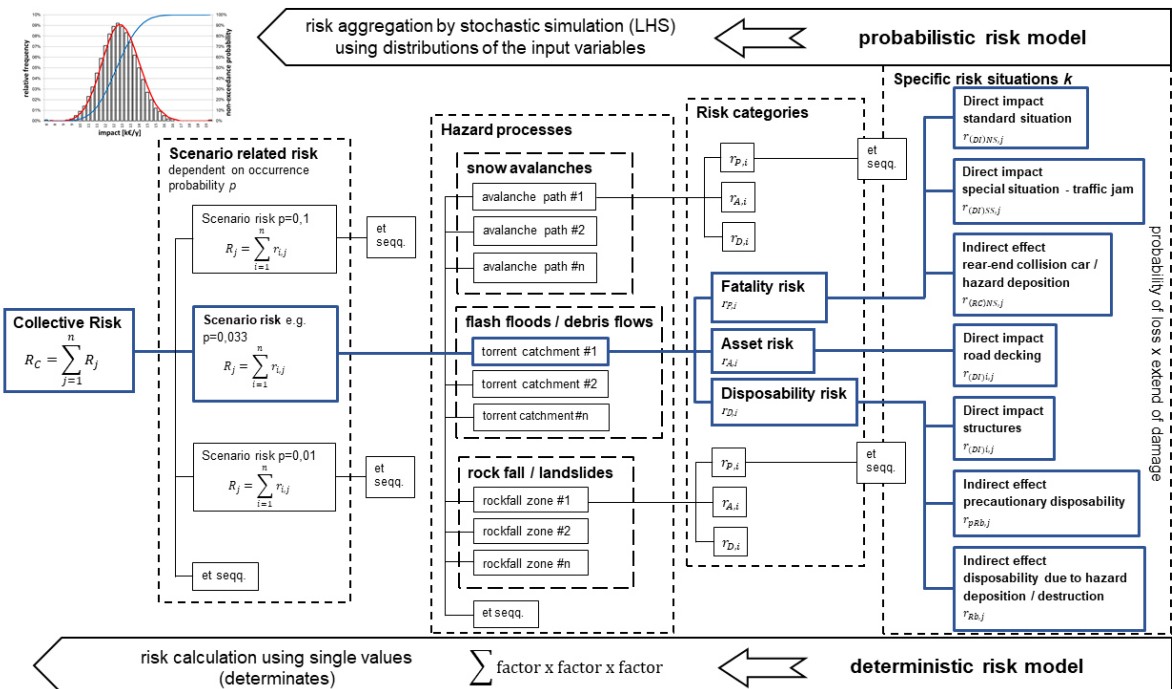

**Figure 1.** Exemplified flow chart for the risk assessment method following the standard approach (deterministic risk
model) from ASTRA (2012) which was supplemented with the probabilistic risk model in present study. In the
deterministic approach each risk variable is addressed with single values and the specific risk situations are summed
up to risk categories for each hazard process class and scenario (probability of occurrence of the hazard process) and

60  finally to the collective risk, whereas the probabilistic setup uses a probability distributions to characterize each risk
61  variable and further aggregates risk by stochastic simulation to the total risk.

## 2 Background

### 2.1 Multi-hazard risk assessment

According to Kappes et al. (2012a), two approaches to multi-hazard risk analysis can be distinguished, a spatially-oriented and a thematically-defined method. While the first aims to include all relevant hazards and associated loss in an area, the latter deals with the influence or interaction of one hazard process on another hazard, frequently addressed as hazards chain or cascading hazards, meaning that the occurrence of one hazard is triggering one or several second-order (successive) hazards. One of the major issues in multi-hazard risk analysis – see Kappes et al. (2012a) for a comprehensive overview – lies in the different process characteristics which lead to challenges for a sound comparison of the resulting risk level among different hazard types due to different reference units. Standardization by a classification scheme for frequency and intensity thresholds of different hazard types resulting in semi-quantitative classes or ranges allows for a comparison among different hazard types, such as shown in Table 2. Therefore, the analysis of risk for transport infrastructure is often focused on an assessment of different hazard types affecting a defined road section rather than on hazard chains or cascades (Schlögl et al., 2019). Following this approach, hazard-specific vulnerability can be assessed either in terms of loss estimates (e.g., Papathoma-Köhle et al., 2011; Fuchs et al., 2019) or in terms of other socioeconomic variables, such as limited access in case of road blockage or interruption (Schlögl et al., 2019). Focusing on the first and neglecting any type of hazard chains, our study demonstrates the application of risk to a specific road section in the Eastern European Alps and shows the sensitivity of the results using deterministic and probabilistic risk approaches.

### 2.2 Deterministic risk concept

Quantitative risk analyses for natural hazards are regularly based on deterministic approaches, and the temporal and spatial occurrence probability of a hazard process with a given magnitude is multiplied by the expected consequences, the latter defined by values at risk times vulnerability (Varnes, 1984; International Organisation for Standardisation, 2009). A universal definition of risk relates the likelihood of an event with the expected consequences, thus manifests risk as a function of hazard times consequences (UNISDR, 2004; ISO, 2009). Depending on the spatial and temporal scale, values at risk include exposed elements, such as buildings (Fuchs et al., 2015, 2017), infrastructure systems (Guikema et al., 2015) and people at risk (Fuchs et al., 2013). These elements at risk are linked to potential loss using vulnerability functions, indices or indicators (Papathoma-Köhle, 2017), and can be expressed in terms of direct and indirect, as well as tangible and intangible loss (Markantonis et al., 2012; Meyer et al., 2013). While direct loss occurs immediately due to the physical impact of the hazard, indirect loss occurs with a certain time lag after an event (Merz et al., 2004, 2010). Furthermore, the distinction between tangible or intangible loss is depending on whether or not the consequences can be assessed in monetary terms. In this context, vulnerability is defined as the degree of loss given to an element of risk as a result from the occurrence of a natural phenomenon of a given intensity, ranging between 0 (no damage) and 1 (total loss) (UNDRO, 1979; Fell et al., 2008; Fuchs, 2009). This definition highlights a physical approach to vulnerability within the domain of natural sciences,

neglecting any societal dimension of risk. However, the expression of vulnerability due to the impact of a threat on
the element at risk considerably differs among hazard types (Papathoma-Köhle et al., 2011).
Using a deterministic approach, the calculation of risk has repeatedly been conceptualised by Eq. (1) (e.g. Fuchs et
al. 2007; Oberndorfer et al. 2007; Bründl et al. 2009) and is dependent on a variety of variables all of which being
subject to uncertainties (Grêt-Regamey and Straub, 2006).

101                 $$R_{i,j} = f\left(p_j,\ p_{i,j},\ A_i,\ v_{i,j}\right) \hspace{4cm} (1)$$

Where $R_{i,j}$ = risk dependent of object $i$ and scenario $j$; $p_j$ = probability of defined scenario $j$; $p_{i,j}$ probability of
exposure of object $i$ to scenario $j$; $A_i$ = value of the object $i$ (the value at risk affected by scenario $j$); $v_{i,j}$ =
vulnerability of the object $i$ in dependence on scenario $j$.
With respect to mountain hazard risk assessment, standardised approaches are available, such as IUGS (1997), Dai et
al. (2002), Bell and Glade (2004), and Fell et al. (2008a, b) for landslides, Bründl et al. (2010) for snow avalanches,
and Bründl (2009) or ASTRA (2012) for a multi-hazard environment. These approaches, however, usually neglect
the inherent uncertainties of involved variables. In particular, they ignore the probability distributions of the variables
(Grêt-Regamey and Straub, 2006) by obtaining the results with constant input parameters, which may lead to bias
(over- and underestimation dependent on the scale of input variables) in the results. Therefore, loss assessment for
natural hazard risk is associated with high uncertainty (Špačková et al., 2014 and Špačková, 2016) and studies
quantifying uncertainties of the expected consequences are underrepresented (Grêt-Regamey and Straub, 2006),
especially regarding natural hazards impacts on roads (Schlögl et al., 2019). For the assessment of an optimal
mitigation strategy for an avalanche-prone road Rheinberger et al. (2009) considers parameter uncertainty by
assuming a joint (symmetric) deviation of ±5 % for all input values to construct a confidence interval for the baseline
risk. The assessment of uncertainty of natural hazard risk is therefore frequently represented by sensitivity analyses
to show the sensitivity through a shift in input values on the results. Thus, the use of confidence intervals allows a
discrete calculation of risk with different model setups. In our study, we quantify the potential uncertainties within
road risk assessment using a stochastic risk assessment approach under consideration of the probability distribution
of input data.
**2.3 Uncertainties within risk assessment**
Since the computation of risk for roads requires a variety of auxiliary calculations, a broad range of input data are
used, such as the spatial and temporal probability of occurrence of specific design events. These auxiliary
calculations subsequently provide variables necessary for risk computation of the respective system under
investigation. Individual contributing variables are often characterized either as mean value of the potential spectrum
from a statistical dataset or, as a consequence of incomplete data, as a single value form expert judgement. Expert
information is frequently processed with semi-quantitative probability classes and therefore subjected to considerable
uncertainties. Consequently, they serve as rough qualitative appraisals encompassing a high degree of uncertainty.
The use of vulnerability parameters or lethality values as a function of process-specific intensities is often based on
incomplete or insufficient statistical data resulting from missing event documentation (Fuchs et al., 2013). As
discussed in Kappes et al. (2012a), Papathoma-Köhle et al. (2011, 2017) and Ciurean et al. (2017) with respect to
mountain hazards, potential sources of uncertainty in vulnerability assessment are independent of the applied

assessment method. The amplitude in data is considerably high in continuous vulnerability curves or functions, but also in discrete (minimum and maximum) vulnerability values referred to as matrices (coefficients), and in indicator-/index-based methods used to calculate the cumulative probability of loss. With regard to the uncertainty in vulnerability matrices, Ciurean et al. (2017) suggested a fully probabilistic simulation in order to quantify the propagation of errors between the different stages of analysis by substituting the range of minimum-maximum values with a probability distribution for each variable in the model.

Grêt-Regamey and Straub (2006) listed potential sources of uncertainties in risk assessment models and classified uncertainties into aleatory and epistemic uncertainties. The first is considered as inherent to a system associated to the natural variability over space and time (Winter et al., 2018) and the variability of underlying random or stochastic processes (Merz and Thieken, 2005, 2009), which cannot be further reduced by an increase in knowledge, information or data. The latter results from incomplete knowledge and can be reduced with an increase of cognition or better information of the system under investigation (Merz and Thieken, 2004, 2009; Grêt-Regamey and Straub, 2006). Particularly referring to deterministic risk analysis, epistemic uncertainty is associated with a lack of knowledge about quantities of fixed but poorly known values (Merz and Thieken, 2009). Špačková (2016) pointed out the importance of interactions (correlations) between uncertainties which may affect the final results, an issue that was also discussed in the framework of multi-hazard risk assessments (Kappes, 2012a, b). Therefore, uncertainties should be included in the analysis by their upper and lower credible limits or by integrating confidence intervals reflecting the incertitude of input data, for an in-depth discussion see e.g. Apel et al. (2004), Merz and Thieken (2004, 2009), Bründl et al. (2009) and Winter et al. (2018).

**2.4 Deterministic vs. probabilistic risk**

Deterministic and probabilistic methods for risk analysis differ significantly in approach. Deterministic methods generally use a defined value (point value) for probability and for the impact (consequence) and consider risk by multiplying the probability of occurrence with the potential consequences. The result is an "expected value" of risk. If multiple risks e.g. with varying frequencies are addressed, the total risk is expressed as the simple sum of single risks resulting in an expected annual average loss. However, information about probability or best and/or worst-case scenarios are often excluded. In particular, the following shortcomings of deterministic approaches can be summarized (Tecklenburg 2003), which in turn leads us to a recommendation of probability-based risk approaches:

- A deterministic method gives equal weight to those risks that have a low probability of occurrence and high impact and to those risks that have a high probability of occurrence and low impact by using a simple multiplication of probability and impact, a topic which is also known as risk aversion effect and is controversially discussed in the literature (e.g., Wachinger et al, 2013; Lechowska, 2018).
- By multiplying the two elements of probability and impact, these values are no longer independent. Therefore, this method is not adequate for aggregation of risks where both probability and impact information need to remain available. Due to multiplication, the only information that remains is the mean value.
- The actual impact will definitely deviate from the deterministic value (i.e., the mean).
- Without the Value at Risk (VaR) information, there is no way to determine how reliable the mean value is and how likely it might be exceeded. The VaR is a measure of risk in economics and describes the probability of

loss within a time unit, which is expressed as a specified quantile of the loss distribution (Cottin and Döhler,
171  2013).

In this context, deterministic systems are perfectly predictable, and the state of the parameters to describe the system
behaviour are fixed (single) values associated with total determinization following an entirely known rule, whereas
probabilistic systems include some degree of uncertainty and the variables/parameters to describe the state of the
system are therefore random (Kirchsteiger, 1999). The variables/parameters in probabilistic systems are described
with probability distributions due to incomplete knowledge, rather than with a discrete single or point value which is
assumed to be totally certain. Probabilistic risk modelling uses stochastic simulation with a defined distribution
function to generate random results within the setting of the boundary conditions. The deterministic variable is
usually included within the input distribution. In Table 1 the two different methods are compared.
**Table 1.** Deterministic versus probabilistic method for risk analysis adjusted and compiled from Sander et al. (2015)
and Kirchsteiger (1999).

|  | Deterministic method | Probabilistic method |
|---|---|---|
| Input | Definition of a single number for consequence as descriptive statements including conservative assumptions expressed by the probability of occurrence multiplied by the impact of the particular hazard. | The probabilistic assessment of risk requires at least one number or – for an entirely probabilistic modelling – a PDF for the probability of occurrence and several values for the impact (e.g., minimum, most likely and maximum) expressed as distribution functions, therefore including uncertainty. |
| Result | A simple mathematical addition to give the aggregated consequence for all risks (point value calculation). This results in an expected consequence for the aggregated risks but does not adequately represent the bandwidth (range) of the aggregated consequences. The deterministic calculation can be supplemented with upper and lower bounds (different model setups) to show the sensitivity of the input on the results using a sensitivity analysis, which are per se separate deterministic calculations. | Simulation methods e.g. Monte Carlo simulation produce a bandwidth (range) of aggregated natural hazards risks as probability distribution based on thousands of coincidental but realistic scenarios (depiction of realistic risk combinations). The method allows an explicit consideration and treatment of all types of reducible uncertainty. |
| Qualification | Results (monetary value or fatality per time unit) are displayed as a single sharp number, which, in itself, does not have an associated probability. | Results are displayed using probability distributions, which allow Value at Risk (VaR) interpretation for each value within the bandwidth (range). |


In our study we present an probabilistic design for loss calculation in order to compute the potential spectrum of
input data with simple distribution functions and further aggregate the intermediate data of exposure situations,
hazard- and scenario-related modules to the probability density function (PDF) of the total collective risk $R_C$ by
means of stochastic simulation (Fig. 1). Consequently, damage induced by natural hazards impact to road
infrastructure as well as to traffic are represented by a range of monetary values as a prognostic distribution of the
expected annual average loss instead of an individual amount.

## 3. Case study

The study area is located in the Eastern European Alps, within the Federal State of Salzburg, Austria (Fig. 2). The case study is a road segment of the federal highway B99 with an overall length of two kilometers ranging from km 52.8 to km 54.8 and is endangered by multiple types of natural hazards. The road segment was chosen to demonstrate the advantages of using probabilistic risk approaches in comparison to traditional deterministic methods. The mountain road under examination is part of a north-south traverse over the main ridge of the Eastern European Alps and is therefore an important regional transit route. Furthermore, the road provides access to the ski resort of Obertauern.

As shown in Fig. 2, the road segment is affected by three avalanche paths, four torrent catchments and one rockfall area. The four torrent catchments have steep alluvial fans on the valley basin. The road segment is located at the base of these fans or the road is slightly notched in the torrential cone and passes the channels either with bridges or with culverts. The rockfall area is situated in the western part of the road segment. Approximately two third of the study area is affected from rock fall processes either as single blocks or by multiple blocks.

The road is frequently used for individual traffic from both sides of the alpine pass. Hence, a mean daily traffic (MDT) of 3,600 cars is observed. This constant frequency represents the standard situation for the potentially exposed elements at risk. However, especially in the winter months the average daily traffic can considerably increase up to an amount of about 7,000 cars. Thus, the traffic data underlies short-term daily and longer-term seasonal fluctuations with peaks up to the double of the mean value. The importance of dynamic risk computation needed for traffic corridors was also discussed earlier by Zischg et al. (2005) and Fuchs et al. (2013) with respect to the spatial-temporal shifts in elements at risk. Besides of the use as a regional transit route, the road is also a central bypass for one of the main transit routes through the Eastern European Alps. Hence, any closure of this main transit route (A10 Tauern motorway) results in a significant increase of daily traffic frequency up to a total of 19,650 cars. The evaluation of the dataset in terms of the bandwidth of the traffic data is shown in Table A6.

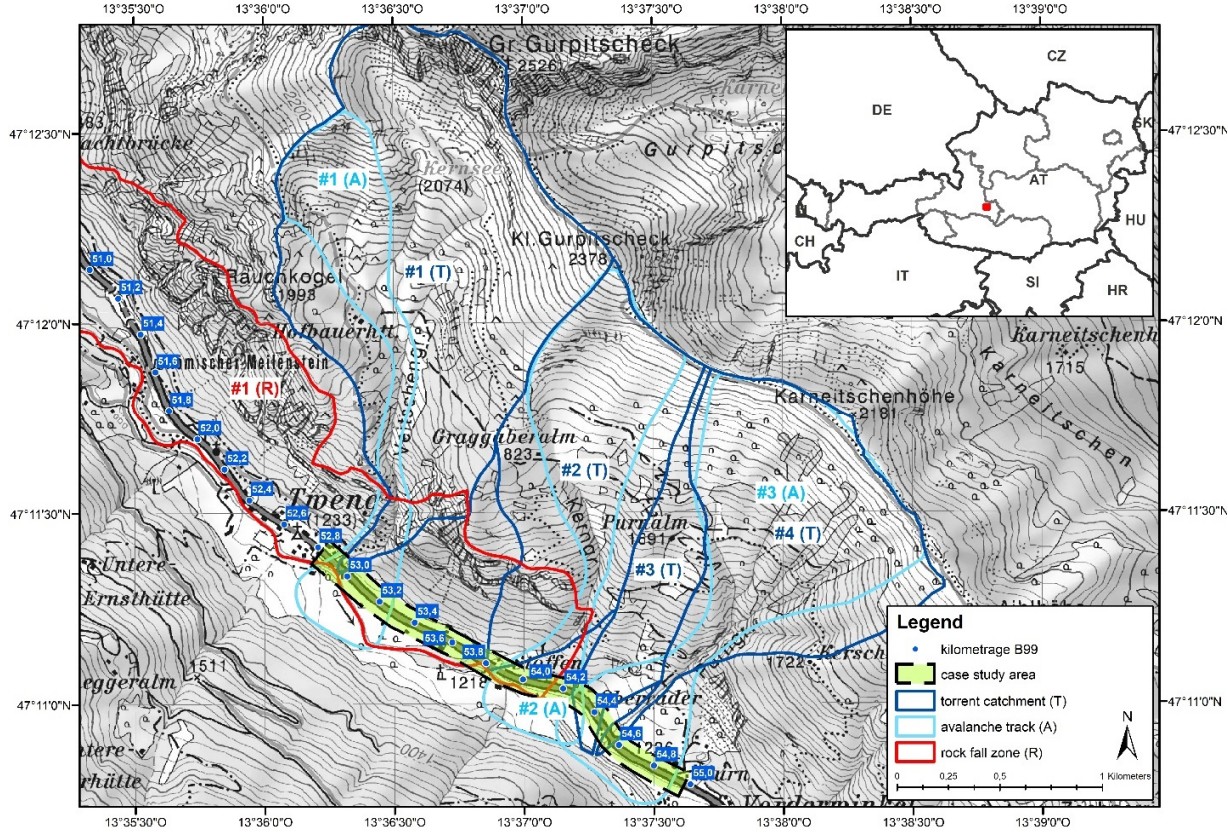

212

**Figure 2.** Overview of the case study area and location of the natural hazards along the road segment (Source base map: © BEV 2020 – Federal Office of Metrology and Surveying, Austria, with permission N2020/69708).

## 4. Methods

### 4.1 Hazard analysis

The hazard analysis was part of technical studies undertaken for the road authority of the Federal State of Salzburg (Geoconsult, 2016; Oberndorfer, 2016). The results regarding the spatial impact of the hazard processes on the elements at risk and the corresponding hazard intensities were used for the loss assessment in this research. The hazard assessment included the steps of hazard disposition analysis to detect potential hazards sources within the perimeter of the road followed by a detailed numerical hazard analysis. Therefore, these analyses considered approaches for hazard-specific impact assessment according to the engineering guidelines of e.g. Bründl (2009), ASTRA (2012) and Bründl et al. (2015) and relevant engineering standards and technical regulations (Austrian Standards Organisation, 2009, 2010, 2017). The physical impact parameters of the hazard processes were calculated using numerical simulation software, such as Flow-2D for flash floods and debris flows (Flow-2D Software, 2017), SamosAT for dense and powder snow avalanches (Sampl, 2007) and Rockyfor3D for rock fall (Dorren, 2012). The hazard analyses were executed without probabilistic calculations; thus, the generated results were integrated as constant input in the risk analysis.

For the multi-hazard purpose three hazard types were evaluated, (1) hydrological hazards (torrential floods, flash
floods, debris flows), (2) geological hazards (rock fall, landslides), and (3) snow avalanches (dense and powder snow
avalanches). For each hazard type, intensity maps for the affected road segment were computed. The intensity maps
specify for a specific hazard scenario the spatial extent of a certain physical impact (e.g., pressure, velocity, or
inundation depth) during a reference period (Bründl et al., 2009). In order to transfer the physical impact to object-
specific vulnerability values for further use in the risk assessment, three process-specific intensity classes were
distinguished (Table 2). These intensity classes were based on the underlying technical guidelines (Bründl, 2009;
ASTRA, 2012; Bründl et al., 2015) and were slightly adapted to comply with the regulatory framework in Austria
(Republik Österreich, 1975, 1976; BMLFUW, 2011). Table 2 represents the intensity classes which correspond to
the affiliated object-specific vulnerability and lethality values (mean damage values) in Tables A7 and A8.
**Table 2.** Process-specific intensity classes with p = pressure, h = height (suffix $h_{ws}$ refers to water and solids), v =
velocity, d = depth and E = energy (compiled and adapted from Bründl (2009), ASTRA (2012) and Republik
Österreich (1975) in conjunction with Republik Österreich (1976) and BMLFUW (2011). The low intensity class for
debris flow has the same intensity indicators than for inundation because it was assumed that low intensity debris
flow events have equal characteristics than hydrological processes.

| Hazard type | Low intensity | Medium intensity | High intensity |
|---|---|---|---|
| Snow avalanche | $1 < p < 3$ kN/m² | $3 < p < 10$ kN/m² | $p > 10$ kN/m² |
| Inundation | $h < 0.5$ m or $v \times h < 0.5$ m²/s | $0.5 < h_{ws} < 1.5$ m or $0.5 < v \times h < 1.5$ m²/s | $h_{ws} > 1.5$ m or $v \times h > 1.5$ m²/s |
| Debris (bed load) deposit | $h_{ws} < 0.5$ m or $v \times h < 0.5$ m²/s | $0.5 < h_s < 0.7$ m or $v < 1$ m/s | $h_s > 0.7$ m and $v > 1.0$ m/s |
| Erosion | -- | $d < 1.5$ m or top edge of the erosion | $d > 1.5$ m or top edge of the erosion |
| Rockfall | $E < 30$ kJ | $30 < E < 300$ kJ | $E > 300$ kJ |

To determine the intensities of individual hazard processes, two different return periods were selected, a 1-in-10-year
and a 1-in-30-year event (probability of occurrence $p_{10} = 0.1$ and $p_{30} = 0.033$). All three snow avalanches can either
develop as powder snow avalanches or as dense flow avalanches, depending on the meteorological and/or snowpack
conditions. Due to the catchment characteristics of the torrents two different indicator processes were assigned for
assessing the hazard effect, depending on the two occurrence intervals. Therefore, the occurrence interval served as a
proxy for the process type since we assumed for the frequently occurring events ($p = 0.1$) the hazard type "flash
floods with sediment transport" and for the medium scale recurrence intervals ($p = 0.033$) debris flow processes.
**4.2 Standard guideline for risk assessment**
The method to calculate road risk for our case study followed the deterministic standard framework of the ASTRA
(2012) guideline for operational road risk assessment. The identification of elements at risk regarding their quantity,
characteristics and value as well as their temporal and spatial variability was assessed through an exposure analysis.
The assessment of the vulnerability of objects (affected road segment, culverts, bridges etc.) and the lethality of
persons was carried out by a consequence analysis to characterize the extent of potential losses. The finally resulting
collective risk $R_C$ (Eqn. 2) as a sum of all hazard types over all object classes and scenarios – under the assumption
that the occurrence of the individual hazards are independent from each other – was expressed in monetary terms per
year as a prognostic value. $R_C$ is therefore defined as the expected annual damage caused by certain hazards and is
frequently used as a risk indicator (Merz et al., 2009; Špačková et al., 2014). Hence, $R_C$ was calculated based on
Eqn. (1) by summing up the partial risk over all scenarios $j$ and objects $i$ (Bründl et al., 2009, Bründl, 2009, ASTRA,
2012, Bründl et al., 2015):
$$R_C = \sum_{j=1}^{n} R_{C,j} \qquad (2)$$

Where $R_{C,j}$ = the total collective risk of scenario $j$ and objects $i$, $R_{C,j} = \sum_{i=1}^{n} r_{i,j}$.
According to the ASTRA (2012) guideline, the collective risk $R_C$ is divided into three main risk groups, (1) risk for
persons $R_P$, (2) property or asset risk $R_A$, and (3) risk of non-operational availability or disposability $R_D$.

### 4.2.1 Risk for persons $R_P$

The risk characterization for persons in terms of the direct impact of a natural hazard on cars was distinguished in a
standard situation for flowing traffic and a situation during a traffic jam, which was seen as specific situation leading
to a significant increase of potentially endangered persons. Additionally, another specific case was also included
representing the rear-end collision either on stagnant cars or on the process depositions on the road in case of the
standard situation. The probability for a rear-end collision depends on the characteristics of the road and is
influenced by a factor of e.g. the visual range, the winding and steepness of the road, the velocity, and traffic density
(ASTRA, 2012). Furthermore, an additional specific scenario was explicitly considered in the case of the road
closure of the main transit route (A10 Tauern motorway) due to the resulting temporal peak of the mean daily traffic.
The statistical mean daily traffic (MDT) was used as mean quantity of persons $N_p$ travelling along the road
(Table A7).
In order to compute $R_P$, the expected annual losses of persons traveling along the road segment under a defined
hazard scenario $j$ was calculated as a combination of the specific damage potential or potential damage extent of
persons and the damage probability of the exposure situation $k$ for persons using the road under investigation. The
potential losses for persons were monetized by the cost for a statistical human life as published by the Austrian
Federal Ministry of Transportation, Innovation and Technology (BMVIT, 2014). The published average national
expenses of road accidents include materially and immaterially costs (body injury, property damage and overhead
expenses) of road accidents and are based on statistical evaluations of the national database as well as on the
willingness to pay approach for human suffering. The monetized costs for a statistical human life equal 3 M€. Thus,
road risk for persons was calculated with three road-specific exposure situations $k$ (Bründl et al., 2009):
1.    Direct impact of the hazard event – standard situation (Eqn. 1A; Table A1)
2.    Direct impact of the hazard event – specific situation due to traffic jam (Eqn. 2A; Table A2)
3.    Indirect effect – rear-end collision (Eqn. 3A; Table A3)
The risk variables to assess $R_P$ are stated in Table A6 for the exposure situations and in Table A7 in the Appendix.

### 4.2.2 Property risk $R_A$

The property risk due to the direct impact of the hazard process on physical assets of the road infrastructure was calculated for each object $i$ and scenario $j$ using Eqn. (4A) with Table A4 under consideration of risk variables in Table A8. The damage probability was assumed to be equal to the frequency of the scenario $j$.

With respect to the potential direct tangible losses within the study area, the physical assets including e.g. the road decking of the street segment, culverts and bridges were expressed by the building costs of the assets calculated from a reference price per unit (Table A8). The physical assets of affected cars were not addressed as this damage type is not included in the standard guideline due to the assumption of an obligatory insurance coverage. The monetized costs refer to replacement costs and reconstruction costs, respectively, instead of depreciated values, which is strongly recommended in risk analysis by Merz et al. (2010) due to the fact that replacement cost systematically overestimates the damage. Since there is a limitation of reliable or even available data on replacement costs, the usage of reconstruction costs is a pragmatic procedure to calculate damage.

### 4.2.3 Risk due to non-operational availability $R_D$

The risk due to non-operational availability can be generally separated into economic losses due to (1) road closure after a hazard event or (2) as a result of precautionary measures for road blockage. The former addresses the mandatory reconditioning of the road and interruption time is depending on the severity of the damage. For our case study, only the precautionary non-operational availability was calculated with Eqn. (5A), Table A5 and variables in Table A9 because the village of Obertauern can be accessed from both directions of the mountain pass road. Therefore, a general accessibility of the village was supposed because it was assumed that events only lead to a road closure on one site of the pass. Potential costs resulting from time delays for necessary detours or e.g. from an increase of environmental or other stresses were neglected. The maximum intensity of the process served as a proxy for the duration of the road closure.

The direct intangible costs for non-operational availability of the road were approximated from statistical data accounting for the business interruption and the loss of profits of the tourism sector in the village of Obertauern due to road closure (see Table A9). The village of Obertauern is a major regional tourism hot spot and therefore the predominant income revenues are based on tourism, thus other business divisions were neglected. Regarding the precautionary expected losses only snow avalanches were included, due to the obligatory legal implementation of a monitoring of a reginal avalanche commission. Thus, a reliable procedure for a road closure could be assumed.

### 4.3 Risk computation

For purpose of computing road risk, the risk Equations 1A to 5A from the standard guideline (ASTRA, 2012), stated in the Appendix in conjunction with Tables A1 to A5, were used without further modification both for the deterministic and for the probabilistic calculation. Hence, the probabilistic setup is based on the same equations as the standard approach, but the variables were addressed with probability distributions instead of single values. In a first step, the deterministic result was computed as a base value for comparison with the results (probability density functions PDFs) of the two diverging probabilistic setups. In a second step, a probabilistic model was integrated into the same calculation setup to consider the band width of the risk-contributing variables. Using this probabilistic

model, the individual risk variables were addressed with two separate probability distributions. The flow chart in Fig. 1 illustrates the risk assessment method and distinguishes between the deterministic and the probabilistic risk model. The diagram exemplarily demonstrates the calculation steps for both model setups. Whereas only the single value of the input data was processed within the standard (deterministic) setup, the probabilistic risk model utilized the bandwidth of each variable denoted in Tables A6 to A9 in the Appendix. These values were either defined from statistical data, expert judgement or from existing literature. The range represents the assumed potential scatter of the variables including a minimum (lower bound $l$), an expected or most likely value ($m$) and a maximum value (upper bund $u$). The deterministic setup was calculated with the expected value, which corresponds in most cases to the recommended input value of the guideline. The choice of the variable range in Tables A6 to A9 in the Appendix is case study specific and cannot be transferred to other studies without careful validation.

### 4.3.1 Probabilistic framework

Within the probabilistic risk modelling setup, the contributing variables for computing the prognostic annual loss were calculated in a stochastic way using their potential range. The probabilistic risk calculation was conducted with the software package RIAAT – Risk Administration and Analysis Tool (RiskConsult, 2016). The probabilistic setup comprised two different and independent calculation runs each with two different distribution functions to characterize the uncertainty of the input variables. Hence, each variable was modelled using either (1) a triangular or three-point distribution (TPD) or (2) a beta-PERT distribution (BPD) within the probabilistic model, which generated two independent probabilistic setups and results. The discrete risk calculation with two different approaches of probability distributions facilitated a comparison of the applicability and the sensitivity of the simple distribution functions on the results. The expected annual monetary losses induced by the three hazard types were aggregated and further compacted to the probability density function (PDF) of the total risk caused by multi-hazard impact. Finally, the two different PDFs from the stochastic risk assessment were compared with the result from the deterministic method to show the potential dynamics in the results.

1. Triangular distribution (TPD)

The triangular distribution derives its statistical properties from the geometry: it is defined by three parameters $l$ for lower bound, $m$ for most likely value (the mode) and $u$ for upper bound. Whereas lower and upper bounds define on both edges the limited bandwidth, the most likely value indicates that values in the middle are more probable than the boundary values, and also allows for the representation of skewness. The TPD is a popular distribution in the risk analysis field (Cottin and Döhler, 2013) for example to reproduce expert estimates. Especially if little or no information about the actual distribution of the parameter or only an estimate of the additional variables to fit the theoretical distribution is feasible, a best possible approximation can be achieved using the TPD. If there is no representative empirical data available as a basis for risk prediction, complex analytical (theoretical) distributions, which are harder to model and communicate, may not represent the reality better than a simple triangular distribution (Sander, 2012).

2. Beta-PERT distribution (BPD)

The beta-PERT distribution (Program Evaluation and Review Technique) is a simplification of the Beta
distribution with the advantage of an easier modelling and application (Sander, 2012). It requires the same three
parameters as a triangular distribution: $l$ for lower bound, $m$ for most likely value (mode) and $u$ for upper bound. In
contrast to the two parametric normal distribution $N(\mu,\sigma)$ – $\mu$ for average and $\sigma$ for standard deviation – the beta-
PERT distribution is limited on the edges and it allows for modelling asymmetric situations. Risk parameters
commonly have a natural boundary, for example vulnerability factors ranging from 0 (no loss) to 1 (total loss).
Therefore, estimating min/max values instead of standard deviation is more realistic or feasible as there is in most
cases no data available to express the mean variation. Moreover, BPD allows for smoother shapes, making it
suitable to model a distribution that is actually an aggregation of several other distributions.
For a given number of risks, each with a probability of occurrence and an individual probability distribution, the
potential number of combinations (scenarios) escalates nonlinear. Especially if dependencies or correlations between
different risks are included and/or numerous partial risks are aggregated to an overall risk the application of
analytical methods have computational restrictions. Stochastic simulations are better suited to work on such complex
models (Tecklenburg, 2003). Therefore, the aggregation of the distributions were calculated by means of Latin
Hypercube sampling (LHS) which is a comparable stochastic simulation technique to Monte-Carlo simulation
(MCS) with the advantage of a faster data processing, a better fitting on the theoretical input distribution and a more
efficiently calculation as fewer iterations are needed to get equally good results (Sander, 2012). LHS consistently
produces values for the distribution's statistics that are nearer to the theoretical values of the input distribution than
MCS. These advantages are possible because the real random numbers used to select samples for the MCS tend to
have local clusters, which are only averaged out for a very large number of draws. Addressing this issue using LHS
can immediately improve the quality of the result by splitting the probability distribution into $n$ intervals of equal
probability, where $n$ is the number of iterations that are to be performed on the model. In the present study, 1,000,000
iterations where performed for every single simulation to get consistent results.
**5. Results and discussion**
In Table 3 the results for each risk group ($R_P$, $R_A$, $R_D$) as well as for the total multi-hazard risk $R_C$ calculated with the
standard deterministic risk approach are shown and compared to those obtained by the two probabilistic setups using
two different probability distributions (TPD and BPD). The results associated with the two distribution functions are
displayed as median value of the PDF to show their deviation to the outcome of the standard approach. Based on our
case study, the road risk over all hazards types and scenarios (multi-hazard risk) with the deterministic approach
results in 76.0 k€/y. The results with the probabilistic approach referring to the median of the PDFs amounts to a
monetary risk of 105.6 k€/y (TPD) and 90.9 k€/y (BPD), respectively. Compared to the standard approach the
median of the PDFs equals an increase of 38 % (BPD) and 19 % (TPD), depending on the choice of probability
distribution to model the uncertainties of the input variables. Focusing on the 95 % percentile (P95) of the results –
non-exceedance probability of 95 %, shown in Fig. 3 – an increase of 79 % (TPD) and 46 % (BDP) to the
deterministic result can be observed. Fig. 3 illustrates, based on the Lorenz curves for the two distributions (TPD and
BPD), the scale of deviation of the total multi-hazard risk $R_C$ within the probabilistic risk modelling and compared to
the standard outcome. The graphs show the potential uncertainties of the risk computation, which can be covered by
a suitable choice of a Value at Risk (VaR) level. For example, with a benchmark of the 95 % quantile (P95), 95 % of
the potential uncertainties within the risk calculation can be covered by using a probabilistic risk assessment
approach. However, a suitable VaR level is depended on the general safety requirement of the system as well as on
the degree of uncertainty of the input variables.

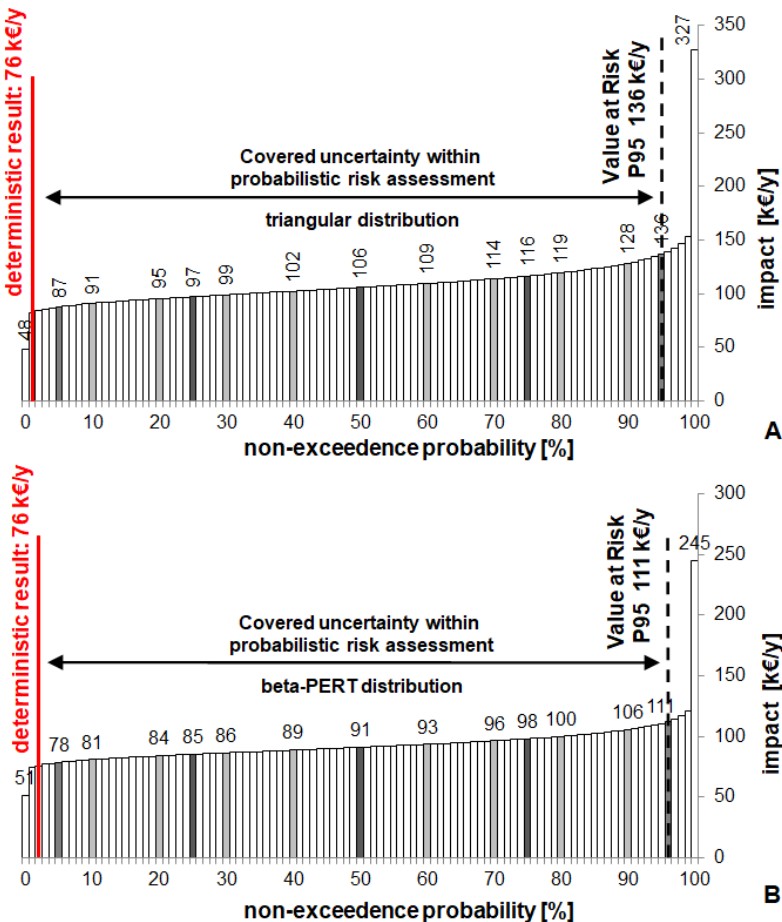


**Figure 3.** Lorenz curves for (A) triangular distribution and (B) beta-PERT distribution showing the scale of
deviation of the total multi-hazard risk $R_C$ within the probabilistic risk modelling and compared to the deterministic
result in k€/y.

Geological hazards (rockfall) contribute with a fraction of 7.8 % to the total risk (or, in absolute numbers, 5.9 k€, see
Table 3) based on the deterministic model, which can be attributed to the relatively small importance in comparison
to the other hazard types in the study area. Hydrological hazards pose the highest risk (50.5 %, or, in absolute
numbers, 38.4 k€/y) previous to avalanche hazards (41.7 %, or, in absolute numbers, 31.7 k€/y). Overall, $R_P$ (44.9 %;
34.1 k€/y) has the highest share on the total multi-hazard risk narrowly followed by $R_A$ (38.9 %; 29.6 k€/y), both
associated to direct damage. The hydrological hazards (predominantly debris flow processes) with a portion of
76.5 % or 26.1 k€/y have a disproportionate high share on $R_P$ due to the high-intensity hazard impact. Similarly, the
semi-empirical lethality factors shown in Table A7 have high values ($\lambda_D = 0.8$) just like the impact of rock fall on
cars with a probability of death of $\lambda_R = 1.0$. Thus, these event types yield in high monetary losses in contrast to snow
avalanches with a lethality factor for high intensity of $\lambda_A = 0.2$. By modelling the hazard-specific lethality with
probability functions a wider scatter can be achieved but the effect still remains due to the heavy weight around the
most likely value $m$. The indirect losses related to $R_D$ with a fraction of 16.3 %, or, in absolute numbers 12.4 k€/y
have a minor portion because this risk group is only relevant for snow avalanches.
**Table 3.** Comparison of the deterministic versus probabilistic results for the three risk categories depending on the
three hazard types and the total collective risk with $R_P$ = risk for persons, $R_A$ asset risk, $R_D$ = disposability risk and $R_C$
= total collective risk with absolute values in k€/y in the first row and as percentage in the second row. For the
probabilistic data, the median value of the triangular $\triangle$ and the beta-PERT $\cap$ distribution functions are displayed.
Note that, risk-based aggregated losses do not equal the sum of the sub-components because probabilistic metrics
such as P50 are not additive. Thus, the computational sum as well as the percentage are slightly different.

| Risk category | | $R_P$ | | | $R_A$ | | | $R_D$ | | | $R_C$ | | |
|---|---|---|---|---|---|---|---|---|---|---|---|---|---|
| Hazard type | Unit | Det. | $\triangle$ | $\cap$ | Det. | $\triangle$ | $\cap$ | Det. | $\triangle$ | $\cap$ | Det. | $\triangle$ | $\cap$ |
| Geological hazards | k€/y | 5.4 | 10.5 | 7.8 | 0.47 | 0.43 | 0.44 | 0 | 0 | 0 | 5.9 | 10.9 | 8.3 |
| | % | 15.8 | 17.0 | 16.3 | 1.6 | 1.4 | 1.5 | 0 | 0 | 0 | 7.8 | 10.3 | 9.1 |
| Hydrological hazards | k€/y | 26.1 | 42.3 | 34.5 | 12.3 | 13.9 | 13.1 | 0 | 0 | 0 | 38.4 | 56.2 | 47.6 |
| | % | 76.5 | 68.3 | 71.9 | 41.6 | 45.6 | 43.5 | 0 | 0 | 0 | 50.5 | 53.2 | 52.4 |
| Avalanche hazards | k€/y | 2.6 | 8.4 | 5.3 | 16.8 | 16.2 | 16.6 | 12.4 | 13.1 | 12.7 | 31.7 | 37.9 | 34.7 |
| | % | 7.6 | 13.6 | 11.0 | 56.8 | 53.1 | 55.1 | 100 | 100 | 100 | 41.7 | 35.9 | 38.2 |
| **Total** | k€/y | **34.1** | **61.9** | **48.0** | **29.6** | **30.5** | **30.1** | **12.4** | **13.1** | **12.7** | **76.0** | **105.6** | **90.9** |
| | % | **44.9** | **58.6** | **52.8** | **38.9** | **28.9** | **33.1** | **16.3** | **12.4** | **14.0** | **100** | **100** | **100** |

The results related to our case study (Table 3 and Fig. 4) show that due to the shape and the mathematical definition
of the distribution the TPD leads to the highest variation in the monetary losses. The boxplots in Fig. 4 display the
results from the probabilistic simulation for the three risk categories ($R_P$, $R_A$, $R_D$) and for the total hazard-specific risk
($R_C$) relating to the three hazard types (Figs. 3 A – C) and for the total multi-hazard collective risk (Fig. 4 D) in
respect of the measures of the central tendency of the PDF. The boxplot diagrams are thereby plotted against the
deterministic value to show its position. The wide range of the distribution in $R_C$ is markedly caused by $R_P$, which
exhibits a broad bandwidth and a right-skewed distribution. Hence, unlike to $R_A$ and $R_D$, the physical injuries
expressed as the economic losses of persons ($R_P$) are responsible for the highest divergence to the standard approach
and show a considerable scatter. The main causes for the striking deviations can be associated to the relatively high
monetary value of persons which was modelled as discrete point value in combination with the fluctuations of the
MDT and the variations of the hazard specific lethality. The monetized costs for a statistical human life equal 3 M€
(Table A7) and is based on a statistical survey of the economic expenses for a road accident in Austria (BMVIT,
2014). Although we ascribe this value to a high degree of uncertainty the valuation of the expenses for a statistical
human life was not attributed to a probability distribution due to the case study-specific fixed governmental
requirements in Austria. The discussion of a monetarily evaluation of a human life is still ongoing across scientific
disciplines using different economic approaches (e.g. Hood, 2017). Furthermore, the lethality factors also correspond
to the high variation of $R_P$ which are seen as very sensitive parameters. Therefore, we encourage further research on
hazard-specific lethality functions for road risk management either based on comprehensive empirical datasets or on
representative hazard impact modelling. Due to the strong effect of $R_P$ on $R_C$ the results have to be carefully
interpreted as they are sensitive to the input variables. Therefore, the values on our case study especially the cost for
human life cannot be directly transferred to other application without a detailed validation and verification of
national regulations.

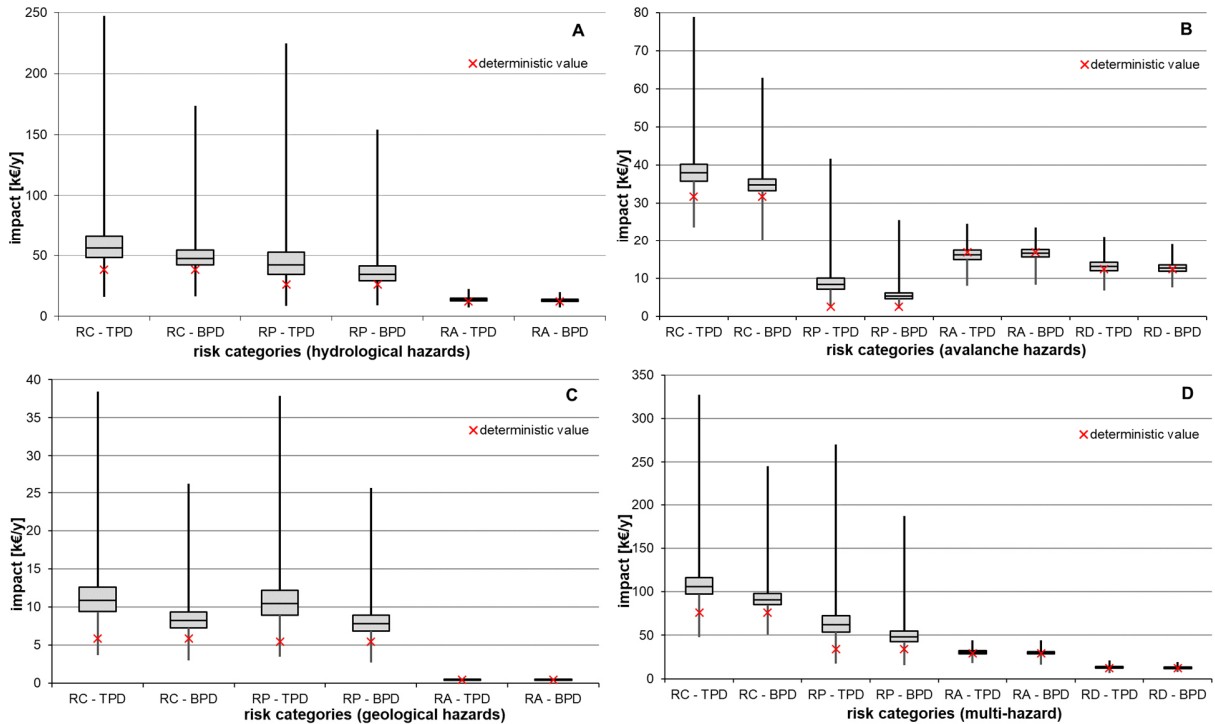


**Figure 4.** Probabilistic results for the three risk categories per hazard type (A = torrent processes, B = snow
avalanches, C = rockfall) and for the total collective risk (D) based on the two distribution functions triangular or
three-point distribution (TPD) and the beta-PERT distribution (BPD) with $R_P$ = risk for persons, $R_A$ asset risk, $R_D$ =
disposability risk and $R_C$ = total collective risk in k€/year.

Apart from $R_P$ where the deterministic result is located below or near the 5 % percentile of both PDFs, $R_A$ and $R_D$ are
mostly within the interquartile range between the 25 % quartile and the median compared to the standard approach
(Fig. 4). In this context, $R_A$ for snow avalanche exceeds the median and is situated between the median and the 75 %
quartile. The effect can be traced back to the left-skewed distribution of the vulnerability factor $v_{B,A}$ for medium
avalanche hazard intensities regarding the object class structures (bridges and culverts) in Table A8. In general, due
to the shape and the mathematical characteristics of the distribution, the BPD leads to a stronger compaction around
the median than the TPD which can be well explained by the properties of the BPD which has, in comparison to the
TPD, a larger weight around the most likely value $m$.

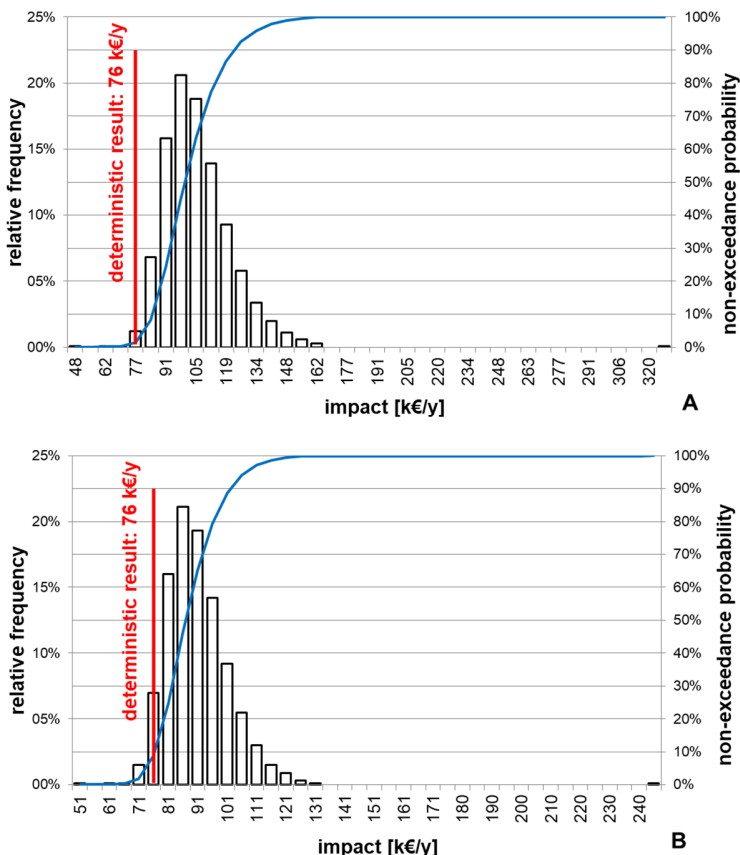

**Figure 5.** Probability density function (PDF) and cumulative distribution function (CDF) for (A) triangular distribution, (B) beta-PERT distribution in k€/y.

In Fig. 5, the PDF and the cumulative distribution function (CDF) are shown for $R_C$ with the two probabilistic model results and the deterministic result. In both cases (TPD and BPD), the deterministic result is situated at the lower edge of the PDF near or under the 5 % percentile. Thus, the deterministic result of our case study covers approximately less than 5 % of the potential band-with of the probability distribution. The TPD has a wide range, whereas the BPD is considerably flattened on the boundary of the amplitude. The results of the two distributions have in common that they are allocated right skewed. In contrast to the location of the median, the deterministic result is on the far-left side of both distribution and is exceeded of more than 95 % of the potential outcome.

## 6 Conclusion

The results based on our case study provide evidence that the monetary risk calculated with a standard deterministic method following the conventional guidelines is lower than applying a probabilistic approach. Thus, without consideration of uncertainty of the input variables risk might be underestimated using the operational standard risk assessment approach for road infrastructure. The mathematical product of the frequency of occurrence and the potential consequences with single values and, in a narrower sense, the multiplication of the partial risk factors in the

second part of the risk equation may lead to a bias in the risk magnitude because the multiplication of the ancillary
calculations generates a theoretical value ignoring the full scope of the total risk.
The far left position of the deterministic value within the PDF of the probabilistic result in our study can be traced
back to fact that the multiplication of two positive symmetrical distributions results in a right-skewed distribution,
because the product of the small numbers at the lower ends of the bandwidths results in much smaller numbers than
the product of the high numbers at the upper ends of the bandwidths. When right-skewed distributions are used as
input and aggregated, the effect of skewness shifts the deterministic value (represented by the most likely value) to
the right side of the resulting distribution. Even if conservative risk values are used in a deterministic setup, a
potential scatter (upper and lower bounds) remains, which leads within a probabilistic calculation through
aggregation of the partial risk elements and sub-results to a right-skewed distribution according to the skewness of
input variables. Since risk values of our study are in most cases asymmetric with primarily positive skews, the
deterministic result migrates during aggregation to the left side of the PDF in Fig. 5. The deterministic risk value is
usually expressed either as a theoretical mean value or as most likely value neglecting the potential distribution
functions of the input data. Thus, the compression of the input values to a single deterministic risk value with total
determination prevents an actual prognosis of reliability that would have been achieved by specifying bandwidths
(Sander, 2012). Furthermore, the simple summation of the scenario related and the object-based risk to receive the
cumulative risk level instead of using probabilistic risk aggregation leads to an underestimation of the final risk.
Hence, the full spectrum of risk cannot be represented with deterministic risk assessment, which may further lead to
biased decisions on risk mitigation.
The Value at Risk (VaR) approach by considering a reliable percentile of the non-exceedance probability e.g. P95 as
shown in Fig. 3 – depending on the desired covering of the risk potential form society, authorities or organizations –
might be an appropriate concept to tackle this challenge. In this context, a higher VaR value implies a higher safety
level for the system under investigation. The final results of risk assessments are subject to uncertainties mainly due
to insufficient data basis of input variables, which can be addressed using a PDF to represent uncertainties involved.
For further decisions on the realization of mitigation measures a high VaR value such as P95 covers these
uncertainties with a defined shortfall probability and thus supports decision makers with more information of road
risk. In turn, as a further practical improvement this benchmark can be compared to the same grade of safety for the
costs of mitigation measures since cost assessments for defence structures are also subject to considerable
uncertainties. Thus, an optimal risk-based design of defence structures might encompass a balance between the same
VaR level both of a probabilistic risk and a probabilistic cost assessment utilizing a cost benefit analysis (CBA).
However, within a probabilistic approach the scale of deviation is dependent on the choice of distribution for
modelling the bandwidth of the variables and the results are sensitive to the defined spectrum of input information
stated in Tables A6 - A9. These variables are case study specific and cannot be directly transferred to other road risk
assessments without careful validation. However, probabilistic risk assessment (PRA) enables a transparent
representation of potential losses due to the explicit consideration of the entire potential bandwidth of the variables
contributing to risk. Since comparable results can be achieved based on predefined values (Bründl et al., 2009), we
still recommend the consideration of the deterministic value as a comparative value to the probabilistic method.

Road risk assessment is usually afflicted to data scarcity; thus, risk operators and practitioners are often dependent on expert appraisals, which are subject to uncertainties. In order to improve data quality, upper and lower values and the expected value can be easily estimated for fitting a simple distribution of the input variables. Even though empirical values such as statistical data are available, a certain degree of uncertainty remains. Therefore, simple distribution functions such as TPD or BPD can adjust the shape of the distribution more conveniently than complex probability distributions, since the required additional parameters to adjust a complex distribution are simple not available. Hence, for a prognostic prediction, risk modelling with complex distributions in contrast to simple techniques cannot be justified if there is a lack of empirical data.

A limitation of our study is that the performance of the probabilistic approach cannot be verified and validated with empirical data, but the results show that the explicit inclusion of epistemic uncertainty leads to a bias in risk magnitude. The probabilistic approach allows quantification of uncertainty, and thus enables decision makers to better assess the quality and validity of the results from road risk assessments. This can facilitate the improvement of road-safety guidelines (for example by implementing a VaR concept), and thus is of particular importance for authorities responsible for operational road-safety, for design engineers and for policy makers due to a general increase of information for optimal decision-making under budget constraints. Furthermore, the paper addresses the second part of the risk concept in terms of the consequence analysis. The results of the hazard analysis serve thereby as a constant input using the physical modelling of the hazard processes without the consideration of probabilistic methods. Thus, the probability of occurrence of the hazard processes was mathematically processed as point value within the probabilistic design since the hazard analyses (with deterministic design events to assess the hazard intensities as a function of the return interval) was part of prior technical studies. Further considerations of a probabilistic modelling of the frequency of the events were outside of the study design and might be addresses in subsequent studies. Therefore, we expect a considerable source of epistemic uncertainty within the hazard analysis which emphasises the necessity for an additional inclusion of probabilistic based hazard analyses in a holistic multi-hazard risk environment. Even though the presented methodology in this study focuses on a road segment exposed to a multi-hazard environment on a local-scale, the approach can easily be transferred to other risk-oriented purposes.

*Acknowledgments:* This work was supported by the Federal State government of Salzburg, Austria, especially from the Geological Service under supervision of L. Fegerl and G. Valentin. The geological hazard analysis was conducted by Geoconsult ZT-GmbH by A. Schober on behalf of the Federal State government of Salzburg which provided the hazard data for the risk analysis. The authors were supported by BOKU Vienna Open Access Publishing Fund.

*Competing interests:* Sven Fuchs is member of the Editorial Board of Natural Hazards and Earth System Sciences.

*Author contribution:* SO initiated the research, was responsible for data collection, literature research, preparation of the manuscript, visualization of the results and performed the risk simulations with additional contribution by PS. PS contributed to additional information on probabilistic risk calculation. SF compiled the background on risk assessment and helped to shape the research, analysis and the manuscript. All authors discussed the results and contributed to the final manuscript.

551     *Data availability:* All risk related data are publicly available (see references throughout the paper as well as in the

552     Appendix).

553

**Appendix**

**Risk equations according to ASTRA (2012) guideline:**

A.   Risk for persons $R_P$

1.   Direct impact of the hazard event – standard situation

$$r_{(DI)NS,j} = p_j \times (1 - p_{Rb}) \times (1 - p_{RbE}) \times p_N \times N_P \times \lambda \times p_{So,j} \times f_L \qquad (1A)$$

**Table A1.** Risk variables and their derivation for the calculation of $R_P$ – direct impact standard situation. [*]The reduction factor considers that not all hazard areas get simultaneously released by the same triggering event. [#]The number of hazard areas for the three hazard types was calculated as discrete values based on field surveys according to the release probability as a function of the event frequency (avalanches $n_{A10} = 6$, $n_{A30} = 7$; torrent processes $n_{T10} = 7$, $n_{T30} = 8$; rockfall $p_{RbE} = 0$ not relevant). [x]The length of the affected street segment is a discrete (single) value according to the results of the hazard analyses.

| Variable | Description | Derivation |
|---|---|---|
| $r_{(DI)NS,j}$ | risk of persons in scenario $j$ (normal situation) | |
| $p_j$ | probability of occurrence of an event (frequency of a scenario $j$) | $p_j = f_j - f_{j+1};\ f_j = \frac{1}{T_j}$ <br> $p_j$ = probability of occurrence of scenario $j$ <br> $f_j$ = frequency of occurrence <br> $T_j$ = return period of scenario $j$ |
| $p_{Rb}$ | probability of precautionary road blockage | |
| $p_{RbE}$ | probability of a road blockage due to an event (road closure due to a previous event of the same hazard type along the road) | $p_{RbE} = \alpha \times \left(1 - \frac{1}{n_H}\right)$ <br> $\alpha$ = reduction factor[*] <br> $n_H$ = number of hazard areas with the same hazard process and triggering mechanism[#] |
| $p_N$ | probability of the standard (normal) situation | $p_N = 1 - p_C$ |
| $p_C$ | probability of a traffic jam (congestion) | $p_C = \left(\frac{n}{365}\right) \times \left(\frac{D}{24}\right)$ <br> $n$ = number of traffic jams per year <br> $D$ = average duration of a traffic jam $[h]$ |
| $N_p$ | number of affected persons | $N_P = N_V \times \beta$ <br> $N_{VN} = \frac{MDT}{v \times 24000} \times l$ = number of vehicles in the standard situation <br> $N_{VJ} = \frac{(\rho_{max} \times l)}{1000}$ = number of vehicles in case of a traffic jam <br> $MDT$ = mean daily traffic <br> $v$ = signalized velocity for cars [km/h] <br> $l$ = length of the street segment [m][x] <br> $\rho_{max}$ = maximum traffic density per lane and kilometer in case of a traffic jam <br> $\beta$ = mean degree of passengers |
| $\lambda$ | lethality factor | Hazard-process and intensity related variable ($\lambda_D, \lambda_F, \lambda_R, \lambda_A$ in table A6) |
| $p_{So,j}$ | spatial occurrence probability of the | for rockfall processes $p_{So,j} = ET \times \frac{d}{w_{HD}}$ |

| | process in the scenario $j$ as proportion of the mean width or area of the process domain in scenario $j$ to the maximum width or area of the potential hazard domain | $ET$ = event type<br>$d$ = mean diameter of the block [m]<br>$w_{HD}$ = width or amplitude of the hazard domain in scenario $j$ |
|---|---|---|
| $f_L$ | factor to differentiate the affected lane | 0,5 = one lane affected<br>1 = whole road (both lanes) affected |

2.    Direct impact of the hazard event – special situation due to traffic jam
$$r_{(DI)SS,j} = p_j \times (1 - p_{Rb}) \times (1 - p_{RbE}) \times p_C \times N_P \times \lambda \times p_{So,j} \times f_L \tag{2A}$$

**Table A2.** Risk of persons in scenario $j$ for the calculation of $R_P$ – direct impact traffic jam. The calculation of the
variables is according to Table A1.

| Variable | Description |
|---|---|
| $r_{(DI)SS,j}$ | risk of persons in scenario $j$ in case of a traffic jam (special situation) |

3.    Indirect effect – Rear-end collision
$$r_{(RC)NS,j} = p_j \times (1 - p_{Rb}) \times (1 - p_{RbE}) \times p_{Rc} \times f_L \times (1 - p_C) \times N_P \times \lambda_{Rc} \tag{3A}$$

**Table A3.** Risk variables and their description for the calculation of $R_P$ – rear-end collision. The calculation of the
residual variables is according to Table A1. [*]A rear-end collision is only valid in case of a standard situation (no
traffic jam). The scenario is not relevant for low intensity hazard events with deposition heights < 0,15 m.

| Variable | Description |
|---|---|
| $r_{(RC)NS,j}$ | risk of persons in scenario $j$ for a rear-end collision in the normal situation[*] |
| $p_{Rc}$ | probability of rear-end collision |
| $\lambda_{Rc}$ | probability of fatality in the case of a rear-end collision |

B.    Property risk $R_A$
$$r_{(DI)i,j} = p_j \times l \times A_i \times v_{i,j} \times p_{So,j} \times f_L \tag{4A}$$

**Table A4.** Risk variables and their description for the calculation of $R_A$ – direct impact. The calculation of the
residual variables is according to Table A1.

| Variable | Description |
|---|---|
| $r_{(DI)i,j}$ | risk of object $i$ in scenario $j$ in terms of a direct impact of the hazard |
| $A_i$ | asset value of object $i$ |
| $v_{i,j}$ | hazard-specific vulnerability of object $i$ in scenario $j$ (in table A7) |
| $l$ | length of the affected road segment |


C.    Risk due to non-operational availability $R_D$
$$r_{Rb,j} = \left(p_j \times f_{Rb} \times \frac{1}{n_H}\right) \times D_{Rb} \times C_{Rb} \tag{5A}$$
**Table A5.** Risk variables and their description for the calculation of $R_D$. The calculation of the residual variables is
according to Table A1.

| Variable | Description |
|---|---|
| $r_{Rb,j}$ | risk of a roadblock in scenario $j$ |
| $f_{Rb}$ | frequency of road blockage |
| $D_{Rb}$ | duration of road blockage depended on the hazard type |
| $C_{Rb}$ | costs of a road blockage |
| $n_H$ | number of hazard areas which are responsible for road closure |

**Risk variables:**
A.    Probability of loss – exposure
**Table A6.** Band width (credible intervals with $l$ - lower bound, $m$ - most likely value and $u$ - upper bound) of the
variables within the probabilistic risk analysis for calculating exposure situations. Units: h for hours, n for numbers,
y for years. [*]Event type 1|5|10 equates to single stone | multiple stones | small scale rockslide.

| Variable | Description | Specification | Unit | $l$ - lower bound | $m$ - most likely value | $u$ - upper bound | Source |
|---|---|---|---|---|---|---|---|
| $p_{Rb}$ | probability of a roadblock | not probable | - | 0 | | | m: ASTRA (2012); l, u: Estimates considering ASTRA class limits |
| | | sparse probable | | 0.05 | 0.1 | 0.5 | |
| | | probable | | 0.1 | 0.5 | 0.9 | |
| | | most likely | | 0.5 | 0.9 | 0.95 | |
| $\alpha$ | reduction factor for $p_{RbE}$ | -- | - | 0.5 | 0.75 | 1 | m: ASTRA (2012); l, u: Expert judgements |
| $n_{B99}$ | number of traffic jams per year | -- | n/y | 0 | 1 | 2 | l, m, u: Expert judgements icw. surveyor of highways (Federal State of Salzburg) |
| $D$ | duration of a traffic jam | -- | h | 0.083 | 0.5 | 2.0 | l, m, u: Expert judgements icw. surveyor of highways (Federal State of Salzburg) |
| $f_{A10}$ | frequency of occurrence special situation A10 | -- | n/y | 5 | 22 | 30 | l, m, u: Statistical evaluation traffic jam database ASFINAG for the year 2015 (min., mean, max. value) |

| | | | | l | m | u | |
|---|---|---|---|---|---|---|---|
| $D_{A10}$ | duration of a special situation A10 | -- | h | 0.5 | 2.65 | 5.0 | l, m, u: Statistical evaluation traffic jam database ASFINAG for the year 2015 (min., mean, max. value) |
| $n_{SS}$ | number of traffic jams in case of a special situation A10 | -- | n | 0 | 4 | 11 | l, m, u: Statistical evaluation traffic jam database ASFINAG for the year 2015 traffic jam events > 0.5h |
| $D_{A10}$ | duration of a traffic jam special situation A10 | -- | h | 0.083 | 1 | 2 | l, m, u: Statistical evaluation traffic jam database ASFINAG for the year 2015 |
| $p_{Rc}$ | Probability of a rear-end collision | improbable | - | 0 | 0.05 | 0.15 | m: ASTRA (2012); l, u: Estimates considering ASTRA class limits |
| | | medium probable | | 0.05 | 0.15 | 0.25 | |
| | | frequent | | 0.15 | 0.25 | 0.35 | |
| ET | event type of rock fall[*] | -- | - | 1 | 5 | 5 | ASTRA (2012) icw. geological expert judgement |

D.    Degree of damage – Risk for persons $R_P$
**Table A7.** Band width (credible intervals *l* - lower bound, *m* - most likely value and *u* - upper bound) of the variables
within the probabilistic risk analysis for calculating $R_P$. Units: h for hours, n for numbers. [*]The monetary value of
person was used as single (point) value as this value is recommended from the Austrian government.

| Variable | Description | Specification | Unit | *l* - lower bound | *m* - most likely value | *u* - upper bound | Source |
|---|---|---|---|---|---|---|---|
| $\lambda_{Rc}$ | probability of fatality in the case of a rear-end collision | -- | - | 0 | 0.0066 | 0.05 | m: ASTRA (2012); l, u: Expert judgements icw. surveyor of highways (Federal State of Salzburg) |
| $\lambda_D$ | lethality for debris flow | low intensity | - | 0 | 0 | 0 | m: ASTRA (2012) and BAFU (2013); l. u: Estimates considering class limits |
| | | medium intensity | | 0 | 0.5005 | 0.7995 | |
| | | strong intensity | | 0.5005 | 0.7995 | 1 | |
| $\lambda_F$ | lethality for dynamic flooding | low intensity | - | 0 | 0 | 0 | m: ASTRA (2012) and BAFU (2013); l. u: Estimates considering class limits |
| | | medium intensity | | 0 | 0.0025 | 0.108 | |
| | | strong intensity | | 0.025 | 0.108 | 0.20 | |
| $\lambda_R$ | lethality for rock fall | low intensity | - | 0 | 0.1 | 0.8 | m: ASTRA (2012) and BAFU (2013) l, u: Estimates considering class limits |
| | | medium intensity | | 0.1 | 0.8 | 1 | |

| Variable | Description | Specification | Unit | l - lower bound | m - most likely value | u - upper bound | Source |
|---|---|---|---|---|---|---|---|
| | | strong intensity | | 0.8 | 1 | 1 | |
| $\lambda_A$ | lethality for avalanche | low intensity | - | 0 | 0.00025 | 0.1 | m: ASTRA (2012) and BAFU (2013); l. u: Estimates considering class limits |
| | | medium intensity | | 0.00025 | 0.1 | 0.2 | |
| | | strong intensity | | 0.1 | 0.2 | 1 | |
| $MDT_{B99}$ | Average daily traffic B99 | -- | n | 3.000 | 3.600 | 7.000 | l, m, u: Traffic counting for the year 2016 (min., mean, max. value) (Federal State of Salzburg) |
| $MDT_{A10}$ | average daily traffic A10 | -- | n | 10.000 | 19.638 | 62.000 | l, m, u: Permanent automatic traffic counting ASFINAG for the year 2016 (min., mean, max. value) |
| $v$ | signalized velocity for cars | free land zone | km/h | 80 | 100 | 120 | m: signalized travel speed; l, u: Expert judgements icw. surveyor of highway (Federal State of Salzburg) |
| | | municipality zone | km/h | 45 | 50 | 60 | |
| | | acceleration / deceleration | km/h | 70 | 80 | 110 | |
| $\rho_{max}$ | maximum traffic density per lane and kilometer in case of a traffic jam | -- | n | 120 | 140 | 145 | m: ASTRA (2012); l, u: Expert judgements icw. surveyor of highway (Federal State of Salzburg) |
| $\beta$ | mean degree of passengers | -- | n | 1 | 1.76 | 5 | m: ASTRA (2012); l, u: Estimates considering one person (driver) and 5 persons in a car. |
| $C_P$ | value (cost) of a person | -- | € | 3,016,194[(*)] | | | BMVIT (2014) for the period 2014-2016 |

E.     Extent of damage – Risk for material assets $R_A$
**Table A8.** Band width (credible intervals $l$ - lower bound, $m$ - most likely value and $u$ - upper bound) of the variables
within the probabilistic risk analysis for calculating $R_A$. [(*)]Base value according to the Federal State of Salzburg:
$l = -20\%$, $u = +10\%$ (right-skewed distribution).

| Variable | Description | Specification | Unit | l - lower bound | m - most likely value | u - upper bound | Source |
|---|---|---|---|---|---|---|---|
| $A_R$ | asset value – construction costs road | -- | €/m | 800 | 850 | 1,000 | l, m, u: Statistical data from Federal State of Salzburg (min., mean, max. value) |

| | | | | l | m | u | |
|---|---|---|---|---|---|---|---|
| $A_B$ | asset value – construction costs bridges (span with 8-10m) | -- | €/m² | 1,350 | 2,200 | 2,400 | l, m, u: Statistical data from Federal State of Salzburg (min., mean, max. value) |
| $A_C$ | asset value – construction costs pipe culverts DN 500-1200 | -- | k€ | 52 | 65 | 71.5[(*)] | m: Statistical data from Federal State of Salzburg l = - 20 %; u = + 10 % (right-skewed distribution) |
| $v_{R,F}$ | vulnerability road dynamic flooding | low intensity | - | 0 | 0.05 | 0.1 | m: ASTRA (2012) and BAFU (2013); l. u: Estimates considering class limits |
| | | medium intensity | | 0.05 | 0.1 | 0.45 | |
| | | strong intensity | | 0.1 | 0.45 | 0.80 | |
| $v_{B,F}$ | vulnerability structures (bridges) dynamic | low intensity | - | 0 | 0.025 | 0.05 | m: ASTRA (2012) and BAFU (2013); l. u: Estimates considering class limits |
| | | medium intensity | | 0.025 | 0.05 | 0.65 | |
| | | strong intensity | | 0.05 | 0.65 | 1 | |
| $v_{R,D}$ | vulnerability road debris flow | low intensity | - | 0 | 0.05 | 0.35 | m: ASTRA (2012) and BAFU (2013); l. u: Estimates considering class limits |
| | | medium intensity | | 0.05 | 0.35 | 0.65 | |
| | | strong intensity | | 0.35 | 0.65 | 1 | |
| $v_{B,D}$ | vulnerability structures (bridges, culvert) debris flow | low intensity | - | 0 | 0.025 | 0.25 | m: ASTRA (2012) and BAFU (2013); l. u: Estimates considering class limits |
| | | medium intensity | | 0.025 | 0.25 | 0.95 | |
| | | strong intensity | | 0.25 | 0.95 | 1 | |
| $v_{R,A}$ | vulnerability road avalanche | low intensity | - | 0 | 0.005 | 0.1 | m: ASTRA (2012) and BAFU (2013); l. u: Estimates considering class limits |
| | | medium intensity | | 0.005 | 0.1 | 0.2 | |
| | | strong intensity | | 0.1 | 0.2 | 0.30 | |
| $v_{B,A}$ | vulnerability structures (bridges, culvert) avalanche | low intensity | - | 0 | 0.005 | 0.7 | m: ASTRA (2012) and BAFU (2013); l. u: Estimates considering class limits |
| | | medium intensity | | 0.005 | 0.7 | 1 | |
| | | strong intensity | | 0.7 | 1 | 1 | |
| $v_{R,R}$ | vulnerability road rock fall | low intensity | - | 0 | 0.1 | 0.5 | m: ASTRA (2012) and BAFU (2013) l, u: Estimates considering class limits |
| | | medium intensity | | 0.1 | 0.5 | 1 | |
| | | strong intensity | | 0.5 | 1 | 1 | |

| | | | | | | | |
|---|---|---|---|---|---|---|---|
| $v_{B,R}$ | vulnerability structures (bridges, culvert) rock fall | low intensity | - | 0 | 0.1 | 0.5 | m: ASTRA (2012) and BAFU (2013) l, u: Estimates considering class limits |
| | | medium intensity | | 0.1 | 0.5 | 1 | |
| | | strong intensity | | 0.5 | 1 | 1 | |

B.    Degree of damage – Risk for operational availability $R_D$
**Table A9.** Band width (credible intervals *l* - lower bound, *m* - most likely value and *u* - upper bound) of the variables
within the probabilistic risk analysis for calculating $R_D$. Units: d for days, n for numbers, y for years.

| Vari-able | Description | Specific-ation | Unit | *l* - lower bound | *m* - most likely value | *u* - upper bound | Source |
|---|---|---|---|---|---|---|---|
| $f_{Rb}$ | frequency of road blockage | -- | n/y | 1 | 2 | 4 | l, m, u: ASTRA (2012) icw. expert judgements (local avalanche commission) |
| $D_{Rb,A10}$ | duration of a precautionary roadblock for avalanche with return interval $T_{10}$ | -- | d | 0.33 | 1 | 2 | l, m, u: ASTRA (2012) icw. expert judgements (local avalanche commission) |
| $D_{Rb,A30}$ | duration of a precautionary roadblock for avalanches with return interval $T_{30}$ | -- | d | 1 | 2 | 3 | l, m, u: ASTRA (2012) icw. expert judgements (local avalanche commission) |
| $C_{Rb,W}$ | expenses of a roadblock during winter season | -- | M€ | 1.245 | 1.557 | 1.868 | m: BMNT (2015) CBA with statistical data of guest-night per hotel category (local tourism agency, 2015) l, u; Range of fluctuation +/- 20 % |

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
