# Peer review of "Multi-hazard risk assessment for roads: Probabilistic versus deterministic approaches"

_Natural Hazards and Earth System Sciences, 2020_

## Referee Comment (RC1) · Anonymous Referee #1 · 8 Apr 2020

General comments:

The paper describes a comprehensive study for 3 main types of risk: risk for persons, property risk and risk for operational availability for mountain hazards (encompassing hydrological hazards, geological hazards and snow avalanches). The paper addresses relevant technical questions within the scope of NHESS. It contains new information about uncertainty of variables used for input to risk assessment for roads to alpine hazards as well as application of probabilistic approaches within transport risk assessment. The methods are clearly described and based on risk equations according to ASTRA (2012) guideline. However, some of the assumptions within the paper

would need more justification (e.g. choice of uncertainty bands for input parameters in risk calculation, i.e. why are upper and lower bands of the input parameters a good approach and where do the applied values for upper and lower bands come from?). The conclusion that the deterministic approach underestimates the risk compared to the probabilistic approach in this study is quite surprising, my experience is that use of deterministic approaches tend to overestimate the risk. The conclusion should be discussed and justified. The abstract contains some unprecise information and some clarifications are suggested. Some of the content should be better structured, to highlight the main contribution of the article and potential applications of the results.

Specific comments:

Suggestion: Extend the start of the introduction; line 25-32. Identify and describe the gaps that this paper is addressing. Introduce a new main section called Background; containing the subsections "Multi-hazard risk assessment", "Deterministic risk concept", "Uncertainties within risk assessment" and "Deterministic vs. probabilistic risk". Include/Move the "Objective" subsection before the suggested "Background section".

Line 12-13: "Due to a variety of variables and data needed for risk computation, a considerable degree of epistemic uncertainty results." : Please clarify this sentence. Why do the need for a variety of variables and data lead to epistemic uncertainty?

Line 14-16: "To overcome this gap, we used a probabilistic approach to express the potential bandwidth of input data with two different distribution functions, taking a mountain road in the Eastern European Alps as case study." a) A bit unprecise formulation, I think. A Probabilistic approach is applied to analyse how the uncertainty in the input data affects the result. The uncertainty in the input data is expressed with a potential band width and two different distribution functions. b) It should also be specified, in general terms for which type of input data uncertainty is included (e.g. exposure, vulnerability and monetary values) and for which they are not included (e.g. hazard

intensities).

Line 16-18: " The risk assessment included the damage potential of road infrastructure and traffic exposed to a multi-hazard environment (torrent processes, snow avalanches, rock fall). : Refer to terms used later in document: Risk for persons, Property risk and Risk for operational availability

Line 21-22: "The results demonstrate that with common deterministic approaches risk is underestimated in comparison to a probabilistic risk modelling setup, mainly due to epistemic uncertainties of the input data." : This conclusion is very surprising. It should be clear that this is only valid for the current study and not generally valid when comparing deterministic and probabilistic results. Usually, conservative values for the input parameters are applied in a deterministic approach to account for the uncertainties – and to provide conservative results. Alternatively, the expected value of the input parameters could be used and the results from the deterministic approach would give the expected value from the probabilistic approach. The validity and explanations for this conclusion should be discussed in the paper.

Line 22-23: "The study provides added value to further develop standardized road safety guidelines and may therefore be of particular importance for road authorities and political decision-makers. : Include in the discussion some thoughts on the application of the results, e.g. how could information about uncertainty in the results be applied within future work to improve the current road safety guidelines.

Line 32: "In contrast, there is still a gap in multi-hazard risk assessments for road infrastructure." a) In which way is this paper also addressing this gap? b) I suggest also to include some introducing text, identifying gaps regarding treatment of uncertainty, to motivate for the coming sub-sections on the topic c) Are there special challenges regarding uncertainties for multi-hazard assessment?

Line 151-159 "Objective" a) The content of the "Objective" subsection should address the scope of the study, referring to the identified gaps described in the introduction, i.e.

**NHESSD**

Interactive
comment

both related to multi hazard assessment and treatment of uncertainties. b) Include: is the multi-hazard risk method in this paper a spatially oriented and a thematically-defined method.

Line 217-219: "Due to the catchment characteristics of the torrents two different indicator processes were assigned for assessing the hazard effect, depending on the two occurrence intervals. Therefore, the occurrence interval served as a proxy for the process type." : I didn't understand this. Could you please clarify/give an example?

Line 213- 224: : Should some of the content be moved to the description of the case study area?

Line 301-302: "These values were either defined from statistical data, expert judgement or from existing literature." : As these values are important for the results; some more documentation on how they were chosen or found should be included, i.e which statistics, literature is applied – or what is the reasoning behind the expert judgment.

Line 335 -337: " In reality, risk parameters commonly have a natural boundary. Therefore, estimating min/max values instead of standard deviation is more realistic or feasible as there is in most cases no data available to express the mean variation. : Justify the use of natural boundaries in this context and what the natural boundaries of risk parameters could be; f.ex. Vulnerability is always between 0 and 1. However; why would there be other natural upper boundaries than 1 in vulnerability; for specific intensities?

Appendix: Tables A6 – A9 : Explain symbols for non-SI units (d, y, n, etc.)

---

## Author Comment (AC1) · 27 Apr 2020

We kindly would like to thank referee #1 for his/her efforts to go through our manuscript and for the insightful and useful comments. Below we chronologically list the questions of the referee – referee comment (RC) and our answers – author comments (AC):

General comments: The paper describes a comprehensive study for 3 main types of risk: risk for persons, property risk and risk for operational availability for mountain hazards (encompassing hydrological hazards, geological hazards and snow avalanches). The paper addresses relevant technical questions within the scope of NHESS. It contains new information about uncertainty of variables used for input to risk assessment

for roads to alpine hazards as well as application of probabilistic approaches within transport risk assessment. The methods are clearly described and based on risk equations according to ASTRA (2012) guideline. However, some of the assumptions within the paper would need more justification (e.g. choice of uncertainty bands for input parameters in risk calculation, i.e. why are upper and lower bands of the input parameters a good approach and where do the applied values for upper and lower bands come from?). The conclusion that the deterministic approach underestimates the risk compared to the probabilistic approach in this study is quite surprising, my experience is that use of deterministic approaches tend to overestimate the risk. The conclusion should be discussed and justified. The abstract contains some unprecise information and some clarifications are suggested. Some of the content should be better structured, to highlight the main contribution of the article and potential applications of the results.

Specific comments:

RC1: Suggestion: Extend the start of the introduction; line 25-32. Identify and describe the gaps that this paper is addressing. Introduce a new main section called Background; containing the subsections "Multi-hazard risk assessment", "Deterministic risk concept", "Uncertainties within risk assessment" and "Deterministic vs. probabilistic risk". Include/Move the "Objective" subsection before the suggested "Background section".

AC1: Thank you for these comments. In a revised version, we will follow your suggestion and structure the article accordingly so that it will become more accessible. Specifically, we will split the current introduction section, introduce a new background section and extend the introduction section, as suggested.

RC2: Line 12-13: "Due to a variety of variables and data needed for risk computation, a considerable degree of epistemic uncertainty results." : Please clarify this sentence. Why do the need for a variety of variables and data lead to epistemic uncertainty?

[Figure]

AC2: Thank you for the comment. We understand that the point we would like to make here can be misunderstood. Hence, in a revised version we will revise this part of the abstract to make it more understandable. The changes foreseen for a revised version will go the following direction:

Abstract. Mountain hazard risk analysis for transport infrastructure is regularly based on deterministic approaches. Standard risk assessment approaches for roads need a variety of variables and data for risk computation, however without considering potential uncertainty in the input data. Consequently, input data needed for risk assessment is normally processed as discrete mean values without scatter, or as an individual deterministic value from expert judgement if no statistical data is available. To overcome this gap, we used a probabilistic approach to analyse the effect of input data uncertainty on the results, taking a mountain road in the Eastern European Alps as case study. The uncertainty of the input data is expressed with potential bandwidths using two different distribution functions. The risk assessment included risk for persons, property risk and risk for non-operational availability exposed to a multi-hazard environment (torrent processes, snow avalanches, rock fall). The study focuses on the epistemic uncertainty of the risk terms (exposure situations, vulnerability factors, monetary values) ignoring potential sources of variation in the hazard analysis. Reliable quantiles of the calculated probability density distributions attributed to the aggregated road risk due to the impact of multiple-mountain hazards were compared to the deterministic results from the standard guidelines on road safety. The results based on our case study demonstrate that with common deterministic approaches risk might be underestimated in comparison to a probabilistic risk modelling setup, mainly due to epistemic uncertainties of the input data. The study provides added value to further develop standardized road safety guidelines and may therefore be of particular importance for road authorities and political decision-makers.

RC3: Line 14-16: "To overcome this gap, we used a probabilistic approach to express the potential bandwidth of input data with two different distribution functions, taking a

mountain road in the Eastern European Alps as case study." a) A bit unprecise formulation, I think. A Probabilistic approach is applied to analyse how the uncertainty in the input data affects the result. The uncertainty in the input data is expressed with a potential band width and two different distribution functions. b) It should also be specified, in general terms for which type of input data uncertainty is included (e.g. exposure, vulnerability and monetary values) and for which they are not included (e.g. hazard C2 NHESSD Interactive comment Printer-friendly version Discussion paper intensities).

AC3: In a revised version we plan to clarify this in the abstract.

RC4: Line 16-18: " The risk assessment included the damage potential of road infrastructure and traffic exposed to a multi-hazard environment (torrent processes, snow avalanches, rock fall). : Refer to terms used later in document: Risk for persons, Property risk and Risk for operational availability

AC4: We will clarify this in the abstract.

RC5: Line 21-22: "The results demonstrate that with common deterministic approaches risk is underestimated in comparison to a probabilistic risk modelling setup, mainly due to epistemic uncertainties of the input data." : This conclusion is very surprising. It should be clear that this is only valid for the current study and not generally valid when comparing deterministic and probabilistic results. Usually, conservative values for the input parameters are applied in a deterministic approach to account for the uncertainties – and to provide conservative results. Alternatively, the expected value of the input parameters could be used and the results from the deterministic approach would give the expected value from the probabilistic approach. The validity and explanations for this conclusion should be discussed in the paper.

AC5: Thank you for this important comment. We will clarify this in the abstract and extend the discussion with the followings paragraph: The multiplication of two positive symmetrical distributions results in a right-skewed distribution, because the product of the small numbers at the lower ends of the bandwidths results in much smaller numbers

than the product of the high numbers at the upper ends of the bandwidths. When right-skewed distributions are used as input and aggregated, the effect of skewness shifts the deterministic value (represented by the most likely value) to the right side of the resulting distribution.

RC6: Line 22-23: "The study provides added value to further develop standardized road safety guidelines and may therefore be of particular importance for road authorities and political decision-makers. : Include in the discussion some thoughts on the application of the results, e.g. how could information about uncertainty in the results be applied within future work to improve the current road safety guidelines.

AC6: Thank you for this comment. Indeed, there is a divide between academia and practical application that can only been closed slowly. We already discussed the applicability for improvement of road safety guidelines in the current version of the manuscript. In a revised version, we will put more focus on this and will provide an additional example for further improvement by implementing a VaR concept to include more information for decision making on road safety issues.

RC7: Line 32: "In contrast, there is still a gap in multi-hazard risk assessments for road infrastructure." a) In which way is this paper also addressing this gap? b) I suggest also to include some introducing text, identifying gaps regarding treatment of uncertainty, to motivate for the coming sub-sections on the topic c) Are there special challenges regarding uncertainties for multi-hazard assessment?

AC8: We will clarify this issue and structure the introduction accordingly.

RC8: Line 151-159 "Objective" a) The content of the "Objective" subsection should address the scope of the study, referring to the identified gaps described in the introduction, i.e. both related to multi hazard assessment and treatment of uncertainties. b) Include: is the multi-hazard risk method in this paper a spatially oriented and a thematicallydefined method.

AC8: Thank you for this comment. As indicated above, we will follow this suggestion and re-structure the manuscript accordingly. We will introduce a background section and extend the introduction section.

RC9: Line 217-219: "Due to the catchment characteristics of the torrents two different indicator processes were assigned for assessing the hazard effect, depending on the two occurrence intervals. Therefore, the occurrence interval served as a proxy for the process type." : I didn't understand this. Could you please clarify/give an example?

AC9: Obviously this sentence is unclear. For clarification we will change this to: "Therefore, the occurrence interval served as a proxy for the process type since we assumed for the frequently occurring events (p = 0.1) the hazard type "flash floods with sediment transport", and for the medium-scale recurrence intervals (p = 0.033) debris flow processes."

RC10: Line 213- 224: : Should some of the content be moved to the description of the case study area?

AC10: Good point. In a revised version of the manuscript we will move the sentence: "As shown in Fig. 2, the road segment is affected by three avalanche paths, four torrent catchments and one rockfall area." Another change will be made with respect to the subsequent statements: "The four torrent catchments have steep alluvial fans on the valley basin. The road segment is located at the base of these fans or the road is slightly notched in the torrential cone and passes the channels either with bridges or with culverts. The rockfall area is situated in the west district of the road segment (Fig. 2). Approximately two third of the study area is affected from rock fall processes either as single blocks or by multiple blocks.". Both will go to the description of the study area.

RC11: Line 301-302: "These values were either defined from statistical data, expert judgement or from existing literature." : As these values are important for the results; some more documentation on how they were chosen or found should be included, i.e which statistics, literature is applied – or what is the reasoning behind the expert

judgment.

AC11: Thank you for your comment. In the Appendix Tables A6 to A9 of the current manuscript version, the source of each variable is quoted. We will further extend the quotations for l / m / u bounds in the source column. The choice of the variable range in Tables A6 to A9 in the Appendix is case study-specific and cannot be transferred to other studies without validation.

RC12: Line 335 -337: " In reality, risk parameters commonly have a natural boundary. Therefore, estimating min/max values instead of standard deviation is more realistic or feasible as there is in most cases no data available to express the mean variation. : Justify the use of natural boundaries in this context and what the natural boundaries of risk parameters could be; f.ex. Vulnerability is always between 0 and 1. However; why would there be other natural upper boundaries than 1 in vulnerability; for specific intensities?

AC12: We will supplement the manuscript with an example of vulnerability factors with boundaries ranging from 0 (no loss) to 1 (total loss).

RC 13: Appendix: Tables A6 – A9 : Explain symbols for non-SI units (d, y, n, etc.)

AC13: The symbols for non-SI-Units will be explained and added in the headings of each table in a revised version of the manuscript.
* * *

---

## Referee Comment (RC2) · Anonymous Referee #2 · 22 Jul 2020

**1  General comments**

The paper compares the results of a deterministic and a probabilistic risk analysis. Input data and parameters of risk analyses are subject to uncertainties mainly due to an insufficient data basis. A sufficient data basis (e.g. lethality values for road accidents due to natural hazards) would allow the derivation of robust data. As such, the paper addresses an important issue because the uncertainties of input data considerably affect the final result of risk analyses, which are often the basis for decisions on the realization of mitigation measures. Practitioners are aware of this issue but have often no explicit numbers at their disposal. Dealing with uncertainties in decisions in practice is an issue under continuing discussion; this paper therefore delivers a valuable contribution to this topic.

The applied method bases on guidelines, tools and papers published some years ago. However, all of them are still in operational use – partly with adaptations –, so the results of this paper refer to actual knowledge and are a valuable contribution to the improvement of these methods and tools.

The topic fits well into the scope of NHESS since it combines scientific research with its application in practice.

The paper is well written, but would benefit from linguistic improvements.

In my opinion, some aspects are not fully clear and could be better explained. Therefore, the paper needs revisions before publication. Some recommendations are given below.

**2  Specific comments**

**2.1  Multi-hazard risk assessment**

The section of multi-hazard risk assessment is very short and therefore only addresses some aspects of this complex topic. I would have expected that this section would show more clearly where are the main gaps and how this paper addresses these gaps. The last sentence targets the difference of results of deterministic vs probabilistic approaches and would therefore fit better in one of the following paragraphs.

**2.2 Deterministic risk concept**

In the paragraph lines 75–83 please explain the inconsistencies you mention (line 79–80). What means inconsistent in this context?

You are right that papers quantifying uncertainties are underrepresented and you cite a paper from 2006. However, meanwhile there are probably much more available. To name only a few, which come to my mind (may be only to show what's missing):

- Rheinberger, C.M., Bründl, M. and Rhyner, J. (2009) Dealing with the White Death: Avalanche Risk Management for Traffic Routes. Risk Analysis 29(1), 76-94.

- Schaub, Y. and Bründl, M. (2010) Zur Sensitivität der Risikoberechnung und Massnahmenbewertung von Naturgefahren. Schweizerische Zeitschrift für das Forstwesen 161(2), 27-35.

- Bründl, M. (2012) EconoMe-Develop - a software tool for assessing natural hazard risk and economic optimisation of mitigation measures. International Snow Science Workshop ISSW, Anchorage, Alaska, pp. 639-643.

If you have done an extensive search on this aspect, it's ok; otherwise I would appreciate to see some papers on uncertainty assessment cited.

**2.3 Deterministic vs. probabilistic risk**

I think, in this section different aspects are discussed, which are not necessarily related to a comparison of deterministic vs probabilistic approaches. I suggest to structure it more clearer. You write in line 126 ". . . a defined value (point value) for probability . . . ".

In my experience, return period intervals, e.g. for a 1 on 10 - 30 years event, are used. Are these point values?

In line 127–129 you write that risk from multiple risks are summed up, which result in an expected average loss. Despite that the term "individual risk" is usually used for the risk an individual person is exposed to (below or above a threshold), this depends how risk is depicted from different processes. Risk can be depicted for each of the processes and for each of the return period intervals (if we speak of return period, which is not the case for non-returning processes such as rockfall). Also the next topic in the bullet point list ("high probability-low consequence . . . ") is not necessarily a topic of a deterministic vs. a probabilistic approach but of weighting, which is known as risk aversion affect (which is controversially discussed especially in the natural hazard community). In the third bullet point, the term "Value at Risk" is mentioned, which should be better explained. Overall, I have the impression that different aspect are mixed and could be structured better.

In **table 1** some things are unclear to me:

**First row:** you write that in a probabilistic assessment of risk one number for the probability of occurrence is required. Deriving the probability of occurrence as part of the hazard analysis is a very critical for a risk analysis if not the most important. In my opinion, the largest uncertainty is probably here (see Schaub and Bründl, 2010, citation above) and a probabilistic method should therefore also handle the uncertainty of the probability of occurrence in order to be really probabilistic. May be you could mention this somewhere in the introduction; its mentioned at the end of the conclusion section.

**Second row:** Mathematical addition in deterministic method: this depends how you aggregate and depict the risks. It is not necessarily the way you describe it here. Upper and lower boundaries are possible.

**Third row:** To my knowledge, the result of a risk analysis is risk, expressed either in monetary terms per time unit, e.g. Euro per year or number of fatalities or injured persons per year. If you differentiate different scenarios, e.g. occurrence probability 0.1, 0.033, etc., you'll get several numbers, which however can be added following conventions (e.g. cumulative-complementary probability).

In **figure 1** the differences between probabilistic and deterministic approach does not become clear to me. The way, risk is calculated is the same, but for the probabilistic approach with a distribution of a parameter, whereas in a deterministic approach, a single parameter is used. This is not clearly shown in the graph. Instead of "Process specific risk classes" you could name the column processes and process areas. What does not come out, how risk from individual process areas are handled (added). In the upper left graph (PDF) the unit "kEuro" for impact represents "risk", right? Then the unit should be "kEuro/year". See also comments below.

2.4   Hazard analysis (section 3.1)

In line 189 you probably mean by "potential hazards" potential release areas which serve as input for the numerical simulation. In line 201, I suggest to replace "expression" by "extent".

In the lines 217–219 it's not clear to me what you want to say. I suggest to rephrase these sentences. In line 223, you might want to replace "west district" by "western part".

2.5   Standard guideline for risk assessment (section 3.2)

I suggest to explain somewhere how you separate the object of risk affected by one or several processes in the different scenarios. What are the objects? Road sections

affected by one single hazard?

In the lines 254–255 you describe the monetization of fatalities. Please briefly mention the approach (I assume by "value of statistical life (VSL)") and the value. Although it can be found in the annex, it would be helpful here.

In the lines 257–259 you give the link to the equations how risk is calculated. Please carefully check the equations for the correct denominations, especially calculation of collective vs. individual risk (see comment below).

2.6   Results and Discussion

I have some problems interpreting the results. Experiences in practice indicate that risk is overestimated compared to real-case events with accidents. In your study you show that deterministic risk analysis underestimates the risk compared to the probabilistic analysis. For me, it becomes not clear why this is the case. The reason could be that the standard value of an input parameter is much too low and the "real" distribution of this input parameter is left skewed (median values are higher than the mean values). But how you know the right distribution?

What would be helpful for the reader is to better explain the meaning of "Value At Risk" (see comment above). Choosing a higher Value-At-Risk-Level (in this case 95% non-exceedance probability) would mean a higher safety level. May be you could write some words more about this concept.

In Figure 3, Table 3, Figure 4 and 5, I see some inconsistencies regarding the units (see also comment above). All numbers which depict risk should must have the unit k per year, so deterministic risk (clearer than "result") and also the "Value At Risk". In Figure 4 and Table 3 I suggest to use the same description of processes. In Table 3 percentages should sum up to 100% (or least close to, which is a problem of rounding of numbers).

As mentioned above, the right-skew in Figure 4 is not clear in relation to the distribution of the input parameters.

For figure 5, I suggest the same scale for both x-axes so that results can be better compared.

At the end of this section, you discuss some consequences of your work for practice. It might be helpful to discuss the consequences of dealing with these uncertainties for practice. Discussions with risk experts reveal that they are aware of uncertainties in input parameters, but it is often not clear how to deal with these results, when uncertainties are explicitly assessed? Communication in practice is very critical in this respect especially to end users such as stakeholders in authorities and communities. What would this mean in regard to the allocation of public money for mitigation measures? Following your argumentation, we could argue that societies in most countries spend too less money for mitigation measures. I think it would be worth to say that your result are the consequences of the chosen distribution of the input values (e.g. upper bounds determined by experts). May be you can add some sentences addressing these aspects.

**3   Technical corrections**

- Line 43: reference to table 2 does not fit here; Please check the order of the table and their numbering.

- Line 186: "The hazard analysis was conducted in technical studies" → "The hazard analysis was part of technical studies"

- Appendix: Please carefully check the equations for the correct denominations, especially calculation of collective vs. individual risk, e.g. Table A1: If you calculate the risk of a person $i$ in scenario $j$, this would the individual risk; therefore

$N_P$ in equation 1A would be 1.

- Table A4: would is the meaning of $l$ in equation 4A?

- Equation 5A: you probably mean $p_j$ instead of $p_i$?

- Table A7: I suggest to use the correction term for $C_P$: it is the value of statistical life (VSL) (?).

- Table A9: variable $C_{Rb,W}$ = expenses?

---

## Author Comment (AC2) · 19 Aug 2020

We kindly would like to thank referee #2 for his/her efforts to evaluate our manuscript and for the insightful and useful comments. Below we chronologically list the questions of the referee – referee comment (RC) and our answers – author comments (AC):

**2 Specific comments**

**2.1 Multi-hazard risk assessment**

(RC1): The section of multi-hazard risk assessment is very short and therefore only addresses some aspects of this complex topic. I would have expected that this section would show more clearly where are the main gaps and how this paper addresses these gaps. The last sentence targets the difference of results of deterministic vs probabilistic approaches and would therefore fit better in one of the following paragraphs.

(AC1): We have deliberately kept this paragraph short and only focused on multi-hazard risk assessment for roads. We cited relevant publications of different hazard processes associated with road risk in the introduction. However, we will restructure the introduction in a revised version of the manuscript so that the overall gap that will be addressed in the manuscript becomes clearer.

**2.2 Deterministic risk concept**

(RC2): In the paragraph lines 75–83 please explain the inconsistencies you mention (line 79–80). What means inconsistent in this context?

(AC2): We will change the term inconsistencies with "*bias*" and will add *"(either over- or underestimation dependent on the scale of input variables)*" to make this sentence clearer.

(RC3): You are right that papers quantifying uncertainties are underrepresented and you cite a paper from 2006. However, meanwhile there are probably much more available. To name only a few, which come to my mind (may be only to show what's missing):

• Rheinberger, C.M., Bründl, M. and Rhyner, J. (2009) Dealing with the White Death: Avalanche Risk Management for Traffic Routes. Risk Analysis 29(1), 76-94.
• Schaub, Y. and Bründl, M. (2010) Zur Sensitivität der Risikoberechnung und Massnahmenbewertung von Naturgefahren. Schweizerische Zeitschrift für das Forstwesen 161(2), 27-35.
• Bründl, M. (2012) EconoMe-Develop - a software tool for assessing natural hazard risk and economic optimisation of mitigation measures. International Snow Science Workshop ISSW, Anchorage, Alaska, pp. 639-643.

If you have done an extensive search on this aspect, it's ok; otherwise I would appreciate to see some papers on uncertainty assessment cited

(AC3): In the current manuscript version, we cited relevant literature in the section "Uncertainties within risk assessment" and we agree that the paper mentioned by the referee is a bit outdated. We will update the text body with newest scholarly works so that the sentence could read as follows: *"Therefore, loss assessment for natural hazard risk is associated with high uncertainty (Špačková et al., 2014 and Špačková, 2016) and studies quantifying uncertainties of the expected consequences are underrepresented (Grêt-Regamey and Straub, 2006), especially regarding natural hazards impacts on roads (Schlögl et al., 2019). For the assessment of an optimal mitigation strategy for an avalanche-prone road Rheinberger et al. (2009) considers parameter uncertainty by assuming a joint (symmetric) deviation of ±5% for all input values to construct a confidence interval for the baseline risk. The assessment of uncertainty of natural hazard risk is therefore frequently represented by sensitivity analyses to show the sensitivity of a shift in input values on the results. Thus, the use of confidence intervals allows a discrete calculation of risk with different model setups. In our study, we quantify the potential uncertainties within road risk assessment using a stochastic risk assessment approach by consideration of the probability distribution of input data".*

**2.3 Deterministic vs. probabilistic risk**

(RC4): I think, in this section different aspects are discussed, which are not necessarily related to a comparison of deterministic vs probabilistic approaches. I suggest to structure it more clearer. You write in line 126 ". . . a defined value (point value) for probability . . . In my experience, return period intervals, e.g. for a 1 on 10 - 30 years event, are used. Are these point values?

(AC4): We thank the referee for this valuable comment. Obviously, the content can be misunderstood, so, in a revised version of the manuscript we will restructure the chapter. The return periods are intervals, but they are mathematically addressed as point values. In our study we only focused on frequent events a 1 in 10 year event and a 1 in 30 year event. In both concepts the probability of occurrence was treated as point values. We totally agree that in a fully probabilistic concept also the probability of occurrence should also be expressed in a probabilistic way. However, since the hazard analysis (with deterministic design events to assess the hazard intensities as a function of the return interval) was part of prior technical studies, further considerations were outside of the study design. This topic might be addressed in a subsequent study.

We will also address this limitation in the conclusion as follows: "*Thus, the probability of occurrence of the hazard processes was mathematically processed as point value within the probabilistic design since the hazard analyses (with deterministic design events to assess the hazard intensities as a function of the return interval) was part of prior technical studies. Further considerations of a probabilistic modelling of the frequency of the events were outside of the study design and might be addresses in subsequent studies*".

(RC5): In line 127–129 you write that risk from multiple risks are summed up, which result in an expected average loss. Despite that the term "individual risk" is usually used for the risk an individual person is exposed to (below or above a threshold), this depends how risk is depicted from different processes. Risk can be depicted for each of the processes and for each of the return period intervals (if we speak of return period, which is not the case for non-returning processes such as rockfall). Also the next topic in the bullet point list ("high probability-low consequence . . . ") is not necessarily a topic of a deterministic vs. a probabilistic approach but of weighting, which is known as risk aversion affect (which is controversially discussed especially in the natural hazard community). In the third bullet point, the term "Value at Risk" is mentioned, which should be better explained. Overall, I have the impression that different aspect are mixed and could be structured better.

(AC5): We agree and will change the term "individual risk" to "single risk" to prevent possible misinterpretation with respect to collective versus individual risk (risk of persons).

We will further extend the first bullet point with the risk aversion discussion as follows:

"*A deterministic method gives equal weight to those risks that have a low probability of occurrence and high impact and to those risks that have a high probability of occurrence and low impact by using a simple multiplication of probability and impact, a topic which is also known as risk aversion affect and controversially discussed in the literature (e.g., Wachinger et al, 2013, Lechowska, 2018)*".

We will also extend the last bullet point with an explanation of the value at risk as follows: "*The VaR is a measure of risk in economics and describes the probability of loss within a time unit, which is expressed as a specified quantile of the loss distribution (Cottin and Döhler, 2013)*".

In table 1 some things are unclear to me:

(RC6): First row: you write that in a probabilistic assessment of risk one number for the probability of occurrence is required. Deriving the probability of occurrence as part of the hazard analysis is a very critical for a risk analysis if not the most important. In my opinion, the largest uncertainty is probably here (see Schaub and Bründl, 2010, citation above) and a probabilistic method should therefore also handle the uncertainty of the probability of occurrence in order to be really probabilistic. May

be you could mention this somewhere in the introduction; its mentioned at the end of the conclusion section.

(AC6): We totally agree with that but we did not model the probability of occurrence in our study in a probabilistic way. In the reversed version of the manuscript we will additionally mention this in the introduction and in the conclusion as you intended to make this more understandable for the readers.

In the introduction we will address this as follows: *"Thus, the probability of occurrence of the hazard event was not assess in a probabilistic way. Since deriving the likelihood of occurrence as part of the hazard analysis is crucial for risk analysis, a high source of uncertainty is attributed to this factor (Schaub and Bründl, 2010)"*.

We will expand this sentence in table 1 as follows: "*The probabilistic assessment of risk requires at least one number or – for an entirely probabilistic modelling – a PDF for the probability of occurrence and several values for the impact (e.g., minimum, most likely and maximum) expressed as distribution functions, therefore including uncertainty*".

(RC7): Second row: Mathematical addition in deterministic method: this depends how you aggregate and depict the risks. It is not necessarily the way you describe it here. Upper and lower boundaries are possible.

(AC7): That's correct, but usually in deterministic risk assessments risk is calculated with standard (single values) and the calculation can bei supplemented with upper and lower bounds to show the sensitivity of the input on the results. This is mostly done by a sensitivity analysis with different model setups which are per se deterministic calculations. This differs from probabilistic analysis where each input variable is treated with a distribution.

We will extend the row with this sentence: "*The deterministic calculation can bei supplemented with upper and lower bounds (different model setups) to show the sensitivity of the input on the results using a sensitivity analysis, which are per se separate deterministic calculations*".

(RC8): Third row: To my knowledge, the result of a risk analysis is risk, expressed either in monetary terms per time unit, e.g. Euro per year or number of fatalities or injured persons per year. If you differentiate different scenarios, e.g. occurrence probability 0.1, 0.033, etc., you'll get several numbers, which however can be added following conventions (e.g. cumulative-complementary probability).

(AC8): We are sorry for this confusion, the referee is right. We will clarify this by including an exemplifying statement such as "*(monetary value or fatality per time unit)*" and will further address this issue in the bullet points.

(RC9): In figure 1 the differences between probabilistic and deterministic approach does not become clear to me. The way, risk is calculated is the same, but for the probabilistic approach with a distribution of a parameter, whereas in a deterministic approach, a single parameter is used. This is not clearly shown in the graph. Instead of "Process specific risk classes" you could name the column processes and process areas. What does not come out, how risk from individual process areas are handled (added). In the upper left graph (PDF) the unit "kEuro" for impact represents "risk", right? Then the unit should be "kEuro/year". See also comments below.

(AC9): Thanks for this important comment. We will change the unit for PDF in the figure to *k€/y* and exchange Process "specific classes" with "*hazard processes*". Moreover, we will explain the flow chart in the figure caption in more detail as follows: *"Figure 1. Exemplified flow chart for the risk assessment method following the standard approach (deterministic risk model) from ASTRA (2012) which was supplemented with the probabilistic risk model in present study. In the deterministic approach each risk variable is addressed with single values and the specific risk situations are summed up to risk categories for each hazard process class and scenario (probability of occurrence of the hazard process) and finally to the collative risk, whereas the*

*probabilistic setup uses a probability distributions to characterize each risk variable and further aggregates risk by stochastic simulation to the total risk".*

2.4 Hazard analysis (section 3.1)

(RC10): In line 189 you probably mean by "potential hazards" potential release areas which serve as input for the numerical simulation. In line 201, I suggest to replace "expression" by "extent".

(AC10): Thank you for this comment. We will change the wording from "potential hazards" to "*potential hazard sources*" and replace "expression" by "*extend*".

(RC11): In the lines 217–219 it's not clear to me what you want to say. I suggest to rephrase these sentences. In line 223, you might want to replace "west district" by "western part".

(AC11): We will rephase the sentences in accordance with the comments of referee #1 to: "*Due to the catchment characteristics of the torrents two different indicator processes were assigned for assessing the hazard effect, depending on the two occurrence intervals. Therefore, the occurrence interval served as a proxy for the process type since we assumed for the frequently occurring events (p = 0.1) the hazard type "flash floods with sediment transport" and for the medium scale recurrence intervals (p = 0.033) debris flow processes."*

We will also replace "west district" into "*western part*" and move this paragraph to the case study section as recommended by referee #1.

2.5 Standard guideline for risk assessment (section 3.2)

(RC12): I suggest to explain somewhere how you separate the object of risk affected by one or several processes in the different scenarios. What are the objects? Road sections C5 affected by one single hazard?

(AC12): Thank you for this comment, we will give explanations of potential affected objects in a revised text, such as *"(affected road segment, culverts, bridges etc.)"*.

(RC13): In the lines 254–255 you describe the monetization of fatalities. Please briefly mention the approach (I assume by "value of statistical life (VSL)") and the value. Although it can be found in the annex, it would be helpful here.

(AC13): We will change the sentence to better focus on the used approach as follows: "*The published average national expenses of road accidents include materially and immaterially costs (body injury, property damage and overhead expenses) of road accidents and are based on statistical evaluations of the national database as well as on the willingness to pay approach for human suffering. The monetized costs for a statistical human life equal 3 M€".*

(RC14): In the lines 257–259 you give the link to the equations how risk is calculated. Please carefully check the equations for the correct denominations, especially calculation of collective vs. individual risk (see comment below).

(AC14): We will check the equation carefully. In our study, however, we focused on the collective risk and excluded the individual risk of highly exposed persons.

2.6 Results and Discussion

(RC15): I have some problems interpreting the results. Experiences in practice indicate that risk is overestimated compared to real-case events with accidents. In your study you show that deterministic risk analysis underestimates the risk compared to the probabilistic analysis. For me, it becomes not clear why this is the case. The reason could be that the standard value of an input parameter is much too low and the "real" distribution of

this input parameter is left skewed (median values are higher than the mean values). But how you know the right distribution?

(AC15): We used two different simple distribution (BPD and TPD) for modelling the bandwidth of each parameter since the actual right distribution of values is not known. We think this is a practical approximation to model a scatter of input data. If more data and research for example for vulnerability or lethality values is available, other more complex distributions may replace these simple distributions.

In the current version of our manuscript we addressed the underestimation of risk in our case study in accordance with the comments of referee #1 as follows: "*Hence, the multiplication of two positive symmetrical distributions results in a right-skewed distribution, because the product of the small numbers at the lower ends of the bandwidths results in much smaller numbers than the product of the high numbers at the upper ends of the bandwidths. When right-skewed distributions are used as input and aggregated, the effect of skewness shifts the deterministic value (represented by the most likely value) to the right side of the resulting distribution.*

*Even if conservative risk values are used in a deterministic setup, a potential scatter (upper and lower bounds) remains, which leads within a probabilistic calculation through aggregation of the partial risk elements and sub-results to a right-skewed distribution according to the skewness of input variables. Since risk values of our study are in most cases asymmetric with primarily positive skews, the deterministic result migrates during aggregation to the left side of the PDF in Fig. 5*".

(RC16): What would be helpful for the reader is to better explain the meaning of "Value At Risk" (see comment above). Choosing a higher Value-At-Risk-Level (in this case 95% nonexceedance probability) would mean a higher safety level. May be you could write some words more about this concept.

(AC16): We will explain the VaR concept (see above) in section 2.4 and complement the VaR in the Conclusion section as recommended by the referee as follows: "*In this context, a higher VaR value implies a higher safety level for the system under investigation*".

(RC17): In Figure 3, Table 3, Figure 4 and 5, I see some inconsistencies regarding the units (see also comment above). All numbers which depict risk should have the unit k per year, so deterministic risk (clearer than "result") and also the "Value At Risk". In Figure 4 and Table 3 I suggest to use the same description of processes. In Table 3 percentages should sum up to 100% (or least close to, which is a problem of rounding of numbers).

(AC17): Thank you for this very important observation. We will change the unites to k€/y in every figure accordingly. In the caption of table 1 we will add to the explanation that "*risk-based aggregated losses do not equal the sum of the sub-components because probabilistic metrics such as P50 are not additive. Thus, the computational sum as well as the percentage are slightly different*".

(RC18): As mentioned above, the right-skew in Figure 4 is not clear in relation to the distribution of the input parameters.

(AC18): Please see our comments to RC15.

(RC19): For figure 5, I suggest the same scale for both x-axes so that results can be better compared.

(AC19): Unfortunately, due to the classes of the frequency plot the scale of the x-axes cannot be changed.

(RC20): At the end of this section, you discuss some consequences of your work for practice. It might be helpful to discuss the consequences of dealing with these uncertainties for practice. Discussions with risk experts reveal that they are aware of uncertainties in input parameters, but it is often not clear how to deal with these results, when uncertainties are explicitly assessed? Communication in practice is very critical in this respect especially to end users such as stakeholders in authorities and communities. What would this mean in regard to the allocation of public money for mitigation measures? Following your argumentation, we could argue that societies in most countries

spend too less money for mitigation measures. I think it would be worth to say that your result are the consequences of the chosen distribution of the input values (e.g. upper bounds determined by experts). May be you can add some sentences addressing these aspects.

(AC20): Thank you very much for this comment. We will address these issues throughout a revised version of the manuscript, and we will particularly extend this discussion at the end of this section, such as:

"*In this context, a higher VaR value implies a higher safety level for the system under investigation. The final results are subject to uncertainties mainly due to insufficient data basis of input variables, which can be addressed using a PDF to represent uncertainties involved. For further decisions on the realization of mitigation measures a high VaR value such as P95 covers these uncertainties with a defined shortfall probability and thus supports decision makers with more information of road risk. In turn, as a further practical improvement this benchmark can be compared to the same grade of safety for the costs of mitigation measures since cost assessments for defence structures are also subject to considerable uncertainties. Thus, an optimal risk-based design of defence structures might encompass a balance between the same VaR level both of a probabilistic risk and a probabilistic cost assessment utilizing a cost benefit analysis (CBA)*".

**Technical corrections**

(RC21): Line 43: reference to table 2 does not fit here; Please check the order of the table and their numbering.

(AC21): We will check this and correct, if necessary.

(RC22): Line 186: "The hazard analysis was conducted in technical studies"*!* "The hazard analysis was part of technical studies".

(AC22): We will change this according to your recommendation.

(RC23): Appendix: Please carefully check the equations for the correct denominations, especially calculation of collective vs. individual risk, e.g. Table A1: If you calculate the risk of a person $i$ in scenario $j$, this would the individual risk; therefore C7 $NP$ in equation 1A would be 1.

(AC23): Thank you very much for this comment. We will fix all headings and descriptions.

(RC24): Table A4: would is the meaning of $l$ in equation 4A?

(AC24): $l$ means the length of the affected road section. We will add the description.

(RC25): Equation 5A: you probably mean $pj$ instead of $pi$?

(AC25): Thank you very much, we will change this expression accordingly.

(RC26): Table A7: I suggest to use the correction term for $CP$: it is the value of statistical life (VSL) (?).

(AC26): We will check and provide explanation in a revised version of the manuscript.

(RC27): Table A9: variable $CRb;W$ = expenses?

(AC27): Thank you very much, we will correct this variable description.